# Empirical Risk Minimization in Non-interactive Local Differential Privacy Revisited *

**Di Wang**　　**Marco Gaboardi**　　**Jinhui Xu**
Department of Computer Science and Engineering
State University of New York at Buffalo
Buffalo, NY, 14260
Email:*{dwang45,gaboardi,jinhui}@buffalo.edu*

## Abstract

In this paper, we revisit the Empirical Risk Minimization problem in the non-interactive local model of differential privacy. In the case of constant or low dimensions ($p \ll n$), we first show that if the loss function is $(\infty, T)$-smooth, we can avoid a dependence of the sample complexity, to achieve error $\alpha$, on the exponential of the dimensionality $p$ with base $1/\alpha$ (*i.e.,* $\alpha^{-p}$), which answers a question in [19]. Our approach is based on polynomial approximation. Then, we propose player-efficient algorithms with 1-bit communication complexity and $O(1)$ computation cost for each player. The error bound is asymptotically the same as the original one. With some additional assumptions, we also give an efficient algorithm for the server. In the case of high dimensions ($n \ll p$), we show that if the loss function is a convex generalized linear function, the error can be bounded by using the Gaussian width of the constrained set, instead of $p$, which improves the one in [19].

## 1 Introduction

Differential privacy [7] has emerged as a rigorous notion for privacy which allows accurate data analysis with a guaranteed bound on the increase in harm for each individual to contribute her data. Methods to guarantee differential privacy have been widely studied, and recently adopted in industry [15, 8].

Two main user models have emerged for differential privacy: the central model and the local one. In the central model, data are managed by a trusted central entity which is responsible for collecting them and for deciding which differentially private data analysis to perform and to release. A classical use case for this model is the one for collecting census data [9]. In the local model, each individual manages his/her proper data and discloses them to a server through some differentially private mechanisms. The server collects the (now private) data of each individual and combines them into a resulting data analysis. A classical application of this model is the one aiming at collecting statistics from user devices like in the case of Google's Chrome browser [8], and Apple's iOS-10 [15, 20].

In the local model, there are two basic kinds of protocols: interactive and non-interactive. Bassily and Smith [2] have recently investigated the power of non-interactive differentially private protocols. This type of protocols seems to be more appealing to real world applications, due to the fact that they can be implemented more easily (*i.e.,* less influenced by the network latency issue). Both Google and Apple use the non-interactive model in their projects [15, 8].

Despite being used in industry, the local model has been much less studied than the central one. Part of the reason for this is that there are intrinsic limitations in what one can do in the local model. As a consequence, many basic questions, that are well studied in the central model, have not been completely understood in the local model, yet.

In this paper, we study differentially private Empirical Risk Minimization in the non-interactive local model. Before showing our contributions and discussing comparisons with previous works, we first discuss our motivations.

**Problem setting [19, 24, 23]** Given a convex, closed and bounded constraint set $\mathcal{C} \subseteq \mathbb{R}^p$, a data universe $\mathcal{D}$, and a loss function $\ell : \mathcal{C} \times \mathcal{D} \mapsto \mathbb{R}$. A dataset $D = \{x_1, x_2 \cdots, x_n\} \in \mathcal{D}^n$ defines an *empirical risk* function: $\hat{L}(\theta; D) = \frac{1}{n} \sum_{i=1}^{n} \ell(\theta, x_i)$. When the inputs are drawn i.i.d from an unknown underlying distribution $\mathcal{P}$ on $\mathcal{D}$, we can also define the *population risk* function: $L_{\mathcal{P}}(\theta) = \mathbb{E}_{D \sim \mathcal{P}^n}[\ell(\theta; D)]$. Now we have the following two kinds of excess risk, one is empirical risk, *i.e.* $\text{Err}_D(\theta_{\text{priv}}) = \hat{L}(\theta_{\text{priv}}; D) - \min_{\theta \in \mathcal{C}} \hat{L}(\theta; D)$, the other one is population risk, *i.e.* $\text{Err}_{\mathcal{P}}(\theta_{\text{priv}}) = L_{\mathcal{P}}(\theta_{\text{priv}}) - \min_{\theta \in \mathcal{C}} L_{\mathcal{P}}(\theta)$.

The problem that we study in this paper is finding $\theta_{\text{priv}} \in \mathcal{C}$ under non-interactive local differential privacy (see Definition 1) which makes the empirical and population excess risk as low as possible. Alternatively, when dimensionality $p$ is a constant or low, we can express this problem in terms of *sample complexity* as finding $n$ as small as possible for achieving $\text{Err}_D \leq \alpha$ and $\text{Err}_{\mathcal{P}} \leq \alpha$, where $\alpha$ is the user specified error tolerance (or simply called error).

**Motivation** Smith et al. [19] prove the following result concerning the problem for general convex 1-Lipschitz loss functions over a bounded constraint set.

**Theorem 1.** Under the assumptions above, there is a non-interactive $\epsilon$-LDP algorithm such that for all distribution $\mathcal{P}$ on $\mathcal{D}$, with probability $1 - \beta$, we have

$$\text{Err}_{\mathcal{P}}(\theta_{\text{priv}}) \leq \tilde{O}\Big(\big(\frac{\sqrt{p} \log^2(1/\beta)}{\epsilon^2 n}\big)^{\frac{1}{p+1}}\Big). \tag{1}$$

A similar result holds for $\text{Err}_D$, with at least $\Omega(n^{\frac{1}{p+1}})$ for both computation and communication complexity for each user. Alternatively, to achieve error $\alpha$, the sample complexity must satisfies $n = \tilde{\Omega}(\sqrt{p} c^p \epsilon^{-2} \alpha^{-(p+1)})$, where $c$ is some constant (approximately 2). More importantly, they also show that generally, the dependence of the sample size over the dimensionality $p$, in the terms $\alpha^{-(p+1)}$ and $c^p$, is unavoidable.

This situation is somehow undesirable: when the dimensionality is high and the target error is low, the dependency on $\alpha^{-(p+1)}$ could make the sample size quite large. However, several results have already shown that for some specific loss functions, the exponential dependency on the dimensionality can be avoided. For example, Smith et al. [19] show that, in the case of linear regression, there is a non-interactive $(\epsilon, \delta)$-LDP algorithm[2] whose sample complexity for achieving error $\alpha$ for the empirical risk is $n = \Omega(p \log(1/\delta)\epsilon^{-2}\alpha^{-2})$. Similarly, Zheng et al. [27] showed that for logistic regression, if the sample complexity satisfies $n > O\big(p(\frac{8r}{\alpha})^{4r \log \log(8r/\alpha)}(\frac{4r}{\epsilon})^{2cr \log(8r/\alpha)+2}(\frac{1}{\alpha^2 \epsilon^2})\big)$, where $c$ and $r$ are independent of $p$, then there is a non-interactive $(\epsilon, \delta)$-LDP algorithm with $\text{Err}_{\mathcal{P}}(\theta_{\text{priv}}) \leq \alpha$.

This propels us to the following natural questions: *i*) Is there any algorithm that has a lower sample complexity than the one in Theorem 1? *ii*) The above discussion indicates that there is a gap between the general case and the case of specific loss functions. Is it possible to introduce some natural conditions on the loss function that guarantee non-interactive $\epsilon$-LDP with sample complexity that is not exponential in the dimensionality $p$? *iii*) The computation and communication costs for each user in the protocols of Smith et al. [19] depend on $n$, which could be high for large datasets. Is it possible to make them independent of $n$? *iv*) The bounds in Smith et al. [19] are not very meaningful in high dimensions. However, in machine learning it is quite common to have high dimensionality, *i.e.*, $n \ll p$. Thus, can we obtain some more meaningful bounds for the high dimensional case? Below we investigate the answer to each question.

**Our Contributions**

1. We first show that by using Bernstein polynomial approximation, it is possible to achieve a non-interactive $\epsilon$-LDP algorithm in constant or low dimensions with the following properties. If the loss function is $(8, T)$-smooth (see Definition 5), with a sample complexity of $n = \tilde{\Omega}\big((c_0 p^{\frac{1}{4}})^p \alpha^{-(2+\frac{p}{2})}\epsilon^{-2}\big)$, the excess empirical risk is ensured to be $\text{Err}_D \leq \alpha$, If the loss function is $(\infty, T)$-smooth, the sample complexity can be further improved to $n \geq \tilde{\Omega}(4^{p(p+1)}D_p^2 p\epsilon^{-2}\alpha^{-4})$, where $D_p$ depends only on $p$. Note that in the first case, the sample complexity is lower than the one in [19] when $\alpha \leq O(\frac{1}{p})$, and in the second case, the sample complexity depends only polynomially on $\alpha^{-1}$, instead of the exponential dependence as in [19]. Furthermore, our algorithm does not assume convexity for the loss function and thus can be applied to non-convex loss functions.

2. Then, we address the efficiency issue, which has only been partially studied before [19]. Following an approach similar to [2], we propose an algorithm for our loss functions which has only 1-bit communication cost and $O(1)$ computation cost for each client, and achieves asymptotically the same error bound as the original one. Additionally, we show also a novel analysis for the server. This shows that if the loss function is convex and Lipschitz and the convex set satisfies some natural conditions, then we have an algorithm which achieves the error bound of $O(p\alpha)$ and runs in polynomial time in $\frac{1}{\alpha}$ (instead of exponential time as in [19]) if the loss function is $(\infty, T)$-smooth.

3. Next, we consider the high dimensional case, and show that if the loss function is a convex generalized linear function, then an $\epsilon$-LDP algorithm is achievable with its risk bound depending only on $n$ and the Gaussian Width of $\mathcal{C}$, which is much smaller than the one in [19]. Particularly, if $\mathcal{C}$ is an $\ell_1$ norm ball or a distribution simplex, the risk bound depends only on $n$ and $\log p$, instead of $p$.

4. Lastly, we show the generality of our technique by applying the polynomial approximation techniques to other problems. We give non-interactive $\epsilon$-LDP algorithms for answering the class of k-way marginals queries and the class of smooth queries, by using different type of polynomials approximations (details are in Supplementary Material).

| Methods | Sample Complexity (omit Poly$(p)$ terms) | Communication Cost (each user) | Computation Cost (each user) | Running time for the server | Assumptions |
|---|---|---|---|---|---|
| Claim 4 in [19] | $\tilde{\Omega}(4^p \alpha^{-(p+2)}\epsilon^{-2})$ | 1 | $O(1)$ | $O\big((\frac{1}{\alpha})^p\big)$ | Lipschitz |
| Theorem 10 in [19] | $\tilde{\Omega}(2^p \alpha^{-(p+1)}\epsilon^{-2})$ | $\Omega(n^{\frac{1}{p+1}})$ | $\Omega(n^{\frac{1}{p+1}})$ | Not Mentioned | Lipschitz and Convex |
| **This Paper** | $\tilde{\Omega}\big((c_0 p^{\frac{1}{4}})^p \alpha^{-(2+\frac{p}{2})}\epsilon^{-2}\big)$ | 1 | $O(1)$ | $O\big((\frac{1}{\alpha})^{\frac{p}{2}}\big)$ | $(8, T)$-smooth |
| **This Paper** | $\tilde{\Omega}(4^{p(p+1)}D_p^2 \epsilon^{-2}\alpha^{-4})$ | 1 | $O(1)$ | $O\big(\text{Poly}(\frac{1}{\alpha})\big)$ | $(\infty, T)$-smooth |

Table 1: Comparisons with existing results in [19] (we assume $p$ is a constant). When the error $\alpha \leq O(\frac{1}{p})$, the sample complexity of $(8, T)$-smooth loss functions is less than the existing result. When the error $\alpha \leq O(\frac{1}{16^p})$, the sample complexity for $(\infty, T)$-smooth loss functions is less than the previous results.

Table 1 shows some comparisons with the results in [19]. Due to the space limit, all proofs and some details of the algorithms are left to the Supplementary Material.

## 2  Related Works

ERM in the local model of differential privacy has been studied in [12, 3, 6, 5, 27, 19, 25]. Kasiviswanathan et al. [12] showed a general equivalence between learning in the local model and learning in the statistical query model. Duchi et al. [6, 5] gave the lower bound $O(\frac{\sqrt{d}}{\epsilon\sqrt{n}})$ and optimal

algorithms for general convex optimization; however, their optimal procedure needs many rounds of interactions. The works that are most related to ours are [27, 19]. Zheng et al. [27] considered some specific loss functions in high dimensions, such as sparse linear regression and kernel ridge regression.

Note that although it also studied a class of loss functions (*i.e.,* Smooth Generalized Linear Loss functions) and used the polynomial approximation approach, the functions investigated in our paper are more general, which include linear regression and logistic regression, and the approximation techniques are quite different. Smith et al. [19] studied general convex loss functions for population excess risk and showed that the dependence on the exponential of the dimensionality is unavoidable. In this paper, we show that such a dependence in the term of $\alpha$ is actually avoidable for a class of loss functions. This even holds for non-convex loss functions, which is quite different from all existing works. Also we study the high dimensional case by using dimension reduction. The polynomial approximation approach has been used under central model in [1, 26, 21, 27] and the dimension reduction has been used in local model in [2, 27].

## 3 Preliminaries

**Differential privacy in the local model.** In LDP, we have a data universe $\mathcal{D}$, $n$ players with each holding a private data record $x_i \in \mathcal{D}$, and a server that is in charge of coordinating the protocol. An LDP protocol proceeds in $T$ rounds. In each round, the server sends a message, which we sometime call a query, to a subset of the players, requesting them to run a particular algorithm. Based on the queries, each player $i$ in the subset selects an algorithm $Q_i$, run it on her data, and sends the output back to the server.

**Definition 1.** [12, 19] An algorithm $Q$ is $\epsilon$-locally differentially private (LDP) if for all pairs $x, x' \in \mathcal{D}$, and for all events $E$ in the output space of $Q$, we have $\Pr[Q(x) \in E] \leq e^\epsilon \Pr[Q(x') \in E]$. A multi-player protocol is $\epsilon$-LDP if for all possible inputs and runs of the protocol, the transcript of player i's interaction with the server is $\epsilon$-LDP. If $T = 1$, we say that the protocol is $\epsilon$ non-interactive LDP.

Since we only consider non-interactive LDP through the paper, we will use LDP as non-interactive LDP below. As an example that will be useful in the sequel, the next lemma shows an $\epsilon$-LDP algorithm for computing 1-dimensional average.

**Lemma 1.** Algorithm 1 is $\epsilon$-LDP. Moreover, if player $i \in [n]$ holds value $v_i \in [0, b]$ and $n > \log \frac{2}{\beta}$ with $0 < \beta < 1$, then, with probability at least $1 - \beta$, the output $a \in \mathbb{R}$ satisfies: $|a - \frac{1}{n} \sum_{i=1}^{n} v_i| \leq \frac{2b\sqrt{\log \frac{2}{\beta}}}{\sqrt{n}\epsilon}$.

---
**Algorithm 1** 1-dim LDP-AVG

1: **Input:** Player $i \in [n]$ holding data $v_i \in [0, b]$, privacy parameter $\epsilon$.
2: **for** Each Player $i$ **do**
3:     Send $z_i = v_i + \mathrm{Lap}(\frac{b}{\epsilon})$
4: **end for**
5: **for** The Server **do**
6:     Output $a = \frac{1}{n} \sum_{i=1}^{n} z_i$.
7: **end for**

---

**Bernstein polynomials and approximation.** We give here some basic definitions that will be used in the sequel; more details can be found in [1, 13, 14].

**Definition 2.** Let $k$ be a positive integer. The Bernstein basis polynomials of degree $k$ are defined as $b_{v,k}(x) = \binom{k}{v} x^v (1-x)^{k-v}$ for $v = 0, \cdots, k$.

**Definition 3.** Let $f : [0, 1] \mapsto \mathbb{R}$ and $k$ be a positive integer. Then, the Bernstein polynomial of $f$ of degree $k$ is defined as $B_k(f; x) = \sum_{v=0}^{k} f(v/k) b_{v,k}(x)$. We denote by $B_k$ the Bernstein operator $B_k(f)(x) = B_k(f, x)$.

**Definition 4.** [14] Let $h$ be a positive integer. The iterated Bernstein operator of order $h$ is defined as the sequence of linear operators $B_k^{(h)} = I - (I - B_k)^h = \sum_{i=1}^{h} \binom{h}{i} (-1)^{i-1} B_k^i$, where $I = B_k^0$ denotes the identity operator and $B_k^i$ is defined as $B_k^i = B_k \circ B_k^{i-1}$. The iterated Bernstein polynomial of order $h$ can be computed as $B_k^{(h)}(f; x) = \sum_{v=0}^{k} f(\frac{v}{k}) b_{v,k}^{(h)}(x)$, where $b_{v,k}^{(h)}(x) = \sum_{i=1}^{h} \binom{h}{i} (-1)^{i-1} B_k^{i-1}(b_{v,k}; x)$.

Iterated Bernstein operator can well approximate multivariate $(h, T)$-smooth functions.

**Definition 5.** [14] Let $h$ be a positive integer and $T > 0$ be a constant. A function $f : [0,1]^p \mapsto \mathbb{R}$ is $(h, T)$-smooth if it is in class $\mathcal{C}^h([0,1]^p)$ and its partial derivatives up to order $h$ are all bounded by $T$. We say it is $(\infty, T)$-smooth, if for every $h \in \mathbb{N}$ it is $(h, T)$-smooth.

**Definition 6.** Assume $f : [0,1]^p \mapsto \mathbb{R}$ and let $k_1, \cdots, k_p, h$ be positive integers. The multivariate iterated Bernstein polynomial of order $h$ at $y = (y_1, \ldots, y_p)$ is defined as:

$$B_{k_1,\ldots,k_p}^{(h)}(f; y) = \sum_{j=1}^{p} \sum_{v_j=0}^{k_j} f(\frac{v_1}{k_1}, \ldots, \frac{v_p}{k_p}) \prod_{i=1}^{p} b_{v_i, k_i}^{(h)}(y_i). \tag{2}$$

We denote $B_k^{(h)} = B_{k_1,\ldots,k_p}^{(h)}(f; y)$ if $k = k_1 = \cdots = k_p$.

**Theorem 2.** [1] If $f : [0,1]^p \mapsto \mathbb{R}$ is a $(2h, T)$-smooth function, then for all positive integers $k$ and $y \in [0,1]^p$, we have $|f(y) - B_k^{(h)}(f; y)| \le O(pTD_h k^{-h})$. Where $D_h$ is a universal constant only related to $h$.

**Our settings** We conclude this section by making explicitly the settings that we will consider throughout the paper. We assume that there is a constraint set $\mathcal{C} \subseteq [0,1]^p$ and for every $x \in \mathcal{D}$ and $\theta \in \mathcal{C}$, $\ell(\cdot, x)$ is well defined on $[0,1]^p$ and $\ell(\theta, x) \in [0,1]$. These closed intervals can be extended to arbitrarily bounded closed intervals. Our settings are similar to the 'Typical Settings' in [19], where $\mathcal{C} \subseteq [0,1]^p$ appears in their Theorem 10, and $\ell(\theta, x) \in [0,1]$ from their 1-Lipschitz requirement and $\|\mathcal{C}\|_2 \le 1$.

## 4 Low Dimensional Case

Definition 6 and Theorem 2 tell us that if we know the value of the empirical risk function, *i.e.* the average of the sum of loss functions, on each of the grid points $(\frac{v_1}{k}, \frac{v_2}{k} \cdots \frac{v_p}{k})$, where $(v_1, \cdots, v_p) \in \mathcal{T} = \{0, 1, \cdots, k\}^p$ for some large $k$, then we can approximate it well. Our main observation is that this can be done in the local model by estimating the average of the sum of loss functions on each of the grid points using Algorithm 1. This is the idea of Algorithm 2.

---

**Algorithm 2** Local Bernstein Mechanism

---

1: **Input:** Player $i \in [n]$ holding data $x_i \in \mathcal{D}$, public loss function $\ell : [0,1]^p \times \mathcal{D} \mapsto [0,1]$, privacy parameter $\epsilon > 0$, and parameter $k$.
2: Construct the grid $\mathcal{T} = \{\frac{v_1}{k}, \ldots, \frac{v_p}{k}\}_{\{v_1,\ldots,v_p\}}$, where $\{v_1, \ldots, v_p\} \in \{0, 1, \cdots, k\}^p$.
3: **for** Each grid point $v = (\frac{v_1}{k}, \ldots, \frac{v_p}{k}) \in \mathcal{T}$ **do**
4:      **for** Each Player $i \in [n]$ **do**
5:          Calculate $\ell(v; x_i)$.
6:      **end for**
7:      Run Algorithm 1 with $\epsilon = \frac{\epsilon}{(k+1)^p}$ and $b = 1$ and denote the output as $\tilde{L}(v; D)$.
8: **end for**
9: **for** The Server **do**
10:      Construct Bernstein polynomial, as in (2), for the perturbed empirical loss $\tilde{L}(v; D)$. Denote $\tilde{L}(\cdot, D)$ the corresponding function.
11:      Compute $\theta_{\text{priv}} = \arg\min_{\theta \in \mathcal{C}} \tilde{L}(\theta; D)$.
12: **end for**

---

**Theorem 3.** For any $\epsilon > 0$ and $0 < \beta < 1$, Algorithm 2 is $\epsilon$-LDP. Assume that the loss function $\ell(\cdot, x)$ is $(2h, T)$-smooth for all $x \in \mathcal{D}$, some positive integer $h$, and constant $T$. If $n, \epsilon$ and $\beta$ satisfy the condition of $n = \Omega\left(\frac{\log \frac{1}{\beta} 4^{p(h+1)}}{\epsilon^2 D_h^2}\right)$, then by setting $k = O\left(\left(\frac{D_h \sqrt{pn}\epsilon}{2^{(h+1)p}\sqrt{\log \frac{1}{\beta}}}\right)^{\frac{1}{h+p}}\right)$, with probability at least $1 - \beta$ we have:

$$\text{Err}_D(\theta_{\text{priv}}) \le \tilde{O}\left(\frac{\log^{\frac{h}{2(h+p)}}(\frac{1}{\beta}) D_h^{\frac{p}{p+h}} p^{\frac{p}{2(h+p)}} 2^{(h+1)p\frac{h}{h+p}}}{n^{\frac{h}{2(h+p)}} \epsilon^{\frac{h}{h+p}}}\right), \tag{3}$$

where $\tilde{O}$ hides the $\log$ and $T$ terms.

From (3) we can see that in order to achieve error $\alpha$, the sample complexity needs to be $n = \tilde{\Omega}(\log \frac{1}{\beta} D_h^{\frac{2p}{h}} p^{\frac{p}{h}} 4^{(h+1)p} \epsilon^{-2} \alpha^{-(2+\frac{2p}{h})})$. This implies the following special cases.

**Corollary 1.** If the loss function $\ell(\cdot, x)$ is $(8, T)$-smooth for all $x \in \mathcal{D}$ and some constant $T$, and $n, \epsilon, \beta, k$ satisfy the condition in Theorem 3 with $h = 4$, then with probability at least $1 - \beta$, the sample complexity to achieve $\alpha$ error is $n = \tilde{O}(\alpha^{-(2+\frac{p}{2})} \epsilon^{-2} (4^5 \sqrt{D_4} p^{\frac{1}{4}})^p)$.

Note that the sample complexity for general convex loss functions in [19] is $n = \tilde{\Omega}(\alpha^{-(p+1)} \epsilon^{-2} 2^p)$, which is considerably worse than ours when $\alpha \leq O(\frac{1}{p})$.

**Corollary 2.** If the loss function $\ell(\cdot, x)$ is $(\infty, T)$-smooth for all $x \in \mathcal{D}$ and some constant $T$, and $n, \epsilon, \beta, k$ satisfy the condition in Theorem 3 with $h = p$, then with probability at least $1 - \beta$, the output $\theta_{\text{priv}}$ of Algorithm 2 satisfies: $\text{Err}_D(\theta_{\text{priv}}) \leq \tilde{O}\left( \frac{\log \frac{1}{\beta}^{\frac{1}{4}} D_p^{\frac{1}{2}} p^{\frac{1}{4}} \sqrt{2}^{(p+1)p}}{n^{\frac{1}{4}} \epsilon^{\frac{1}{2}}} \right)$, where $\tilde{O}$ hides the log and $T$ terms. So, to achieve error $\alpha$, with probability at least $1 - \beta$, we have sample complexity:

$$n = \tilde{\Omega}\left( \max\{ 4^{p(p+1)} \log(\frac{1}{\beta}) D_p^2 p \epsilon^{-2} \alpha^{-4}, \frac{\log \frac{1}{\beta} 4^{p(p+1)}}{\epsilon^2 D_p^2} \} \right). \tag{4}$$

It is worth noticing that from (3) we can see that when the term $\frac{h}{p}$ grows, the term $\alpha$ decreases. Thus, for loss functions that are $(\infty, T)$-smooth, we can get a smaller dependency than the term $\alpha^{-4}$ in (4). For example, if we take $h = 2p$, then the sample complexity is $n = \Omega(\max\{ c_2^{p^2} \log \frac{1}{\beta} D_{2p} \sqrt{p} \epsilon^{-2} \alpha^{-3}, \frac{\log \frac{1}{\beta} c^{p^2}}{\epsilon^2 D_{2p}^2} \})$ for some constants $c, c_2$. When $h \to \infty$, the dependency on the error becomes $\alpha^{-2}$, which is the optimal bound, even for convex functions.

Our analysis of the empirical excess risk does not use the convexity assumption. While this gives a bound which is not optimal, even for $p = 1$, it also says that our result holds for non-convex loss functions and constrained domain set, as long as they are smooth enough.

From (4), we can see that our sample complexity is lower than the one in [19] when $\alpha \leq O(\frac{1}{16^p})$. Note that to achieve the best performance for the ERM problem in low dimensional space, quite often the error is set to be extremely small, *e.g.*, $\alpha = 10^{-10} \sim 10^{-14}$[10].

Using the convexity assumption of the loss function, and a lemma in [18], we can also give a bound on the population excess risk, details are in Supplementary Material.

Corollary 1 and 2 provide answers to our motivative questions. That is, for loss functions which are $(8, T)$-smooth, we can obtain a lower sample complexity; if they are $(\infty, T)$-smooth, there is an $\epsilon$-LDP algorithm for the empirical and population excess risks achieving error $\alpha$ with sample complexity which is independent from the dimensionality $p$ in the term $\alpha$. This result does not contradict the results in Smith et al. [19]. Indeed, the example used to show the unavoidable dependency between the sample complexity and $\alpha^{-\Omega(p)}$, to achieve the $\alpha$ error, is actually non-smooth.

However, in our result of $(\infty, T)$-smooth case, like in the one by Smith et al. [19], there is still a dependency of the sample complexity in the term $c^p$, for some constant $c$. There is still the question about what condition would allow a sample complexity independent from this term. We leave this question for future research and we focus instead on the efficiency and further applications of our method.

## 5 More Efficient Algorithms

Algorithm 2 has computational time and communication complexity for each player which is exponential in the dimensionality. This is clearly problematic for every realistic practical application. For this reason, in this section, we study more efficient algorithms. In order for convenience, in this part we only focus on the case of $(\infty, T)$-smooth loss functions, it can be easily extended to general cases.

Consider the following lemma, showing an $\epsilon$-LDP algorithm for computing $p$-dimensional average (notice the extra conditions on $n$ and $p$ compared with Lemma 1).

**Lemma 2.** [16] Consider player $i \in [n]$ holding data $v_i \in \mathbb{R}^p$ with coordinate between $0$ and $b$. Then for $0 < \beta < 1$, $0 < \epsilon$ such that $n \geq 8p \log(\frac{8p}{\beta})$ and $\sqrt{n} \geq \frac{12}{\epsilon} \sqrt{\log \frac{32}{\beta}}$, there is an $\epsilon$-LDP

algorithm, LDP-AVG, with probability at least $1 - \beta$, the output $a \in \mathbb{R}^p$ satisfying: $\max_{j \in [p]} |a_j - \frac{1}{n} \sum_{i=1}^n [v_i]_j| \leq O(\frac{bp}{\sqrt{n}\epsilon} \sqrt{\log \frac{p}{\beta}})^3$. Moreover, the computation cost for each user is $O(1)$.

By using this lemma and by discretizing the grid with some interval steps, we can design an algorithm which requires $O(1)$ computation time and $O(\log n)$-bits communication per player (see Supplementary Material). However, we would like to do even better and obtain constant communication complexity. Instead of discretizing the grid, we apply a technique, firstly proposed by Bassily and Smith [2], which permits to transform any 'sampling resilient' $\epsilon$-LDP protocol into a protocol with 1-bit communication complexity. Roughly speaking, a protocol is sampling resilient if its output on any dataset $S$ can be approximated well by its output on a random subset of half of the players.

Since our algorithm only uses the LDP-AVG protocol, we can show that it is indeed sampling resilient. Inspired by this result, we propose Algorithm 3 and obtain the following theorem.

**Theorem 4.** For $0 < \epsilon \leq \ln 2$ and $0 < \beta < 1$, Algorithm 3 is $\epsilon$-LDP. If the loss function $\ell(\cdot, x)$ is $(\infty, T)$-smooth for all $x \in \mathcal{D}$ and $n = \Omega(\max\{\frac{\log \frac{1}{\beta} 4^{p(p+1)}}{\epsilon^2 D_p^2}, p(k+1)^p \log(k+1), \frac{1}{\epsilon^2} \log \frac{1}{\beta}\})$, then by setting $k = O\big((\frac{D_p \sqrt{pn}\epsilon}{2^{(p+1)p} \sqrt{\log \frac{1}{\beta}}})^{\frac{1}{2p}}\big)$, the results in Corollary 2 hold with probability at least $1 - 4\beta$.

Moreover, for each player the time complexity is $O(1)$, and the communication complexity is 1-bit.

---

**Algorithm 3** Player-Efficient Local Bernstein Mechanism with 1-bit communication per player

---

1: **Input:** Player $i \in [n]$ holding data $x_i \in \mathcal{D}$, public loss function $\ell : [0,1]^p \times \mathcal{D} \mapsto [0,1]$, privacy parameter $\epsilon \leq \ln 2$, and parameter $k$.
2: **Preprocessing:**
3: Generate $n$ independent public strings
4: $y_1 = \text{Lap}(\frac{1}{\epsilon}), \cdots, y_n = \text{Lap}(\frac{1}{\epsilon})$.
5: Construct the grid $\mathcal{T} = \{\frac{v_1}{k}, \ldots, \frac{v_p}{k}\}_{\{v_1, \ldots, v_p\}}$, where $\{v_1, \ldots, v_p\} \in \{0, 1, \cdots, k\}^p$.
6: Partition randomly $[n]$ into $d = (k+1)^p$ subsets $I_1, I_2, \cdots, I_d$, and associate each $I_j$ to a grid point $\mathcal{T}(j) \in \mathcal{T}$.
7: **for** Each Player $i \in [n]$ **do**
8:     Find $I_l$ such that $i \in I_l$. Calculate $v_i = \ell(\mathcal{T}(l); x_i)$.
9:     Compute $p_i = \frac{1}{2} \frac{\Pr[v_i + \text{Lap}(\frac{1}{\epsilon}) = y_i]}{\Pr[\text{Lap}(\frac{1}{\epsilon}) = y_i]}$
10:     Sample a bit $b_i$ from Bernoulli$(p_i)$ and send it to the server.
11: **end for**
12: **for** The Server **do**
13:     **for** $i = 1 \cdots n$ **do**
14:         Check if $b_i = 1$, set $\tilde{z}_i = y_i$, otherwise $\tilde{z}_i = 0$.
15:     **end for**
16:     **for** each $l \in [d]$ **do**
17:         Compute $v_\ell = \frac{n}{|I_l|} \sum_{i \in I_\ell} \tilde{z}_i$
18:         Denote the corresponding grid point $(\frac{v_1}{k}, \ldots, \frac{v_p}{k}) \in \mathcal{T}$ of $I_l$, then denote $\hat{L}((\frac{v_1}{k}, \cdots, \frac{v_p}{k}); D) = v_l$.
19:     **end for**
20:     Construct Bernstein polynomial for the perturbed empirical loss $\tilde{L}$ as in Algorithm 2. Denote $\tilde{L}(\cdot, D)$ the corresponding function.
21:     Compute $\theta_{\text{priv}} = \arg \min_{\theta \in \mathcal{C}} \tilde{L}(\theta; D)$.
22: **end for**

---

Now we study the algorithm from the server's complexity perspective. The polynomial construction time complexity is $O(n)$, where the most inefficient part is finding $\theta_{\text{priv}} = \arg \min_{\theta \in \mathcal{C}} \tilde{L}(\theta, D)$. In fact, this function may be non-convex; but unlike general non-convex functions, it can be $\alpha$-uniformly approximated by a convex function $\hat{L}(\cdot; D)$ if the loss function is convex (by the proof of

Theorem 3), although we do not have access to it. Thus, we can see this problem as an instance of Approximately-Convex Optimization, which has been studied recently by Risteski and Li [17].

**Definition 7.** [17] We say that a convex set $\mathcal{C}$ is $\mu$-well-conditioned for $\mu \geq 1$, if there exists a function $F : \mathbb{R}^p \mapsto \mathbb{R}$ such that $\mathcal{C} = \{x | F(x) \leq 0\}$ and for every $x \in \partial K : \frac{\|\nabla^2 F(x)\|_2}{\|\nabla F(x)\|_2} \leq \mu$.

**Lemma 3** (Theorem 3.2 in [17]). Let $\epsilon, \Delta$ be two real numbers such that $\Delta \leq \max\{\frac{\epsilon^2}{\mu\sqrt{p}}, \frac{\epsilon}{p}\} \times \frac{1}{16348}$. Then, there exists an algorithm $\mathcal{A}$ such that for any given $\Delta$-approximate convex function $\tilde{f}$ over a $\mu$-well-conditioned convex set $\mathcal{C} \subseteq \mathbb{R}^p$ of diameter 1 (that is, there exists a 1-Lipschitz convex function $f : \mathcal{C} \mapsto \mathbb{R}$ such that for every $x \in \mathcal{C}, |f(x) - \tilde{f}(x)| \leq \Delta$), $\mathcal{A}$ returns a point $\tilde{x} \in \mathcal{C}$ with probability at least $1 - \delta$ in time $\text{Poly}(p, \frac{1}{\epsilon}, \log\frac{1}{\delta})$ and with the following guarantee $\tilde{f}(\tilde{x}) \leq \min_{x \in \mathcal{C}} \tilde{f}(x) + \epsilon$.

Based on Lemma 3 (for $\tilde{L}(\theta; D)$) and Corollary 2, and taking $\epsilon = O(p\alpha)$, we have the following.

**Theorem 5.** Under the conditions in Corollary 2, and assuming that $n = \tilde{\Omega}(4^{p(p+1)} \log(1/\beta) D_p^2 p\epsilon^{-2}\alpha^{-4})$, that the loss function $\ell(\cdot, x)$ is 1-Lipschitz and convex for every $x \in \mathcal{D}$, that the constraint set $\mathcal{C}$ is convex and $\|\mathcal{C}\|_2 \leq 1$, and satisfies $\mu$-well-condition property (see Definition 7), if the error $\alpha$ satisfies $\alpha \leq C\frac{\mu}{p\sqrt{p}}$ for some universal constant $C$, then there is an algorithm $\mathcal{A}$ which runs in $\text{Poly}(n, \frac{1}{\alpha}, \log\frac{1}{\beta})$ time[4] for the server, and with probability $1 - 2\beta$ the output $\tilde{\theta}_{\text{priv}}$ of $\mathcal{A}$ satisfies $\tilde{L}(\tilde{\theta}_{\text{priv}}; D) \leq \min_{\theta \in \mathcal{C}} \tilde{L}(\theta; D) + O(p\alpha)$, which means that $\text{Err}_D(\tilde{\theta}_{\text{priv}}) \leq O(p\alpha)$.

Combining with Theorem 4, 5 and Corollary 2, and taking $\alpha = \frac{\alpha}{p}$, we have our final result:

**Theorem 6.** Under the conditions of Corollary 2, Theorem 4 and 5, and for any $C\frac{\mu}{\sqrt{p}} > \alpha > 0$, if we further set $n = \tilde{\Omega}(4^{p(p+1)} \log(1/\beta) D_p^2 p^5 \epsilon^{-2}\alpha^{-4})$, then there is an $\epsilon$-LDP algorithm, with $O(1)$ running time and 1-bit communication per player, and $\text{Poly}(\frac{1}{\alpha}, \log\frac{1}{\beta})$ running time for the server. Furthermore, with probability at least $1 - 5\beta$, the output $\tilde{\theta}_{\text{priv}}$ satisfies $\text{Err}_D(\tilde{\theta}_{\text{priv}}) \leq O(\alpha)$.

## 6 High Dimensional Case

In previous sections, $p$ is assumed to be either constant or low. In this section, we present a general method for a family of loss functions, called generalized linear functions, in high dimensions.

A function $\ell(w, x)$ is called a Generalized Linear Function (GLF) [18] if $\ell(w, x) = f(\langle w, y\rangle, z)$ for $x = (y, z)$, where $y \in \mathbb{R}^p$ is the data and $z$ is the label. GLF is a rather general family of functions, including many frequently encountered loss functions like logistic regression, hinge loss, linear regression, etc. We assume that the dataset satisfies the conditions of $\|y_i\| \leq 1$ and $\|z_i\| \leq 1$ for all $i \in [n]$. Also, $f$ is assumed to be 1-Lipschitz, convex, $\|\mathcal{C}\|_2 \leq 1$ and isotropic [5].

Our algorithm is inspired by the one in [11]. We first conduct a dimension reduction for the whole dataset. That is, $D' = \{(\Phi y_1, z_1), \cdots, (\Phi y_n, z_n)\}$, where $\Phi \in \mathbb{R}^{m \times p}$. Then, we run a modified version of the algorithm in [19] (since the algorithm in [19] assumes $\log n \geq p$). After obtaining the private estimator $\bar{w} \in \mathbb{R}^m$, we use a compressive sensing technique (by solving an optimization problem [22]) to recover $w^{\text{priv}} \in \mathbb{R}^p$. Our method is based on the following lemma in [4].

**Lemma 4.** Let $\tilde{\Phi} \in \mathbb{R}^{m \times p}$ be an random matrix, whose rows are i.i.d mean-zero, isotropic, sub-Gaussian random variable in $\mathbb{R}^d$ with $\psi = \|\Phi_i\|_{\psi_2}$. Let $\Phi = \frac{1}{\sqrt{m}}\tilde{\Phi}$ and $S$ be a set of points in $R^d$. Then, there is a constant $C > 0$ such that for any $0 < \gamma, \beta < 1$. $\Pr[\sup_{a \in S} |\|\Phi a\|^2 - \|a\|^2 \leq \gamma\|a\|^2] \leq \beta$, provided that $m \geq \frac{C\psi^4}{\gamma^2} \max\{\mathcal{G}_S^2, \log(1/\beta)\}$.

**Theorem 7.** Under the assumption above. For any $\epsilon \leq \frac{1}{4}$, Algorithm 4 is $O(\epsilon)$-LDP. Moreover, setting $m = \Theta(\frac{\psi^4(\mathcal{G}_{\mathcal{C}} + \sqrt{\log n})^2 \log(n/\beta)}{\gamma^2})$, where $\gamma = \Theta(\frac{\psi\sqrt{(\mathcal{G}_{\mathcal{C}} + \sqrt{\log n})}\log(1/\beta)\sqrt[4]{\log(n/\beta)}}{\sqrt{n}\epsilon})$, then with

**Algorithm 4** DR-ERM-LDP

---
1: **Input:** Player $i \in [n]$ holding data $x_i = (y_i, z_i) \in \mathcal{D}$, where $\|y_i\| \le 1$, privacy parameter $\epsilon$.
2: The server generates an random sub-Gaussian matrix $\Phi \in \mathbb{R}^{m \times p}$ in Lemma 3, and sends the seed of this random matrix to all players.
3: **for** Each Player $i$ **do**
4:     Calculate $x_i' = (\Phi y_i, z_i)$
5:     Run the modified $\epsilon$-local DP algorithm of [19](see Supplementary Material for more details) for $D' = \{x_i'\}_{i=1}^n$ with constrained set $\mathcal{C} = \Phi\mathcal{C}$ and loss function $f$. The server get the output as $\bar{w} \in \mathbb{R}^m$.
6: **end for**
7: The server solves the following problem $w^{\mathrm{priv}} = \arg\min_{w \in \mathbb{R}^p} \|w\|_{\mathcal{C}}$, subject to $\Phi w = \bar{w}$.

---

probability at least $1 - \beta$, $\mathrm{Err}_D(w^{\mathrm{priv}}) = \tilde{O}\left(\left(\frac{\log(1/\beta)\psi\sqrt{(\mathcal{G}_{\mathcal{C}}+\sqrt{\log n})}\sqrt[4]{\log(n/\beta)}}{\sqrt{n}\epsilon}\right)^{\frac{1}{1+m}}\right)$, where $\psi$ is the subgaussian norm of the distribution of $\Phi$, and $\mathcal{G}_{\mathcal{C}}$ is the Gaussian width of $\mathcal{C}$.

**Corollary 3.** If $\Phi$ is a standard Gaussian random matrix, $\mathcal{C}$ is the $\ell_1$ norm ball $B_1^p$ or the distribution simplex in $\mathbb{R}^p$, and $n \ll p \le e^{cn}$ for some constant $c$, then the bound in Theorem 7 is just $\tilde{O}\left(\left(\frac{\log(1/\beta)\sqrt[4]{\log p}\sqrt[4]{\log(n/\beta)}}{\sqrt{n}\epsilon}\right)^{\frac{1}{1+m}}\right)$, where $m = O(n\epsilon^2 \log p \sqrt{\log(n/\beta)})$. Note that the bound in this case is always better than the one in Theorem 1, since it is always $O(1)$.

## Footnotes

*This research was supported in part by the National Science Foundation (NSF) under Grant No. CCF-1422324, CCF-1716400, CCF-1718220 and CNS-1565365.

[2] Although, these two results are formulated for non-interactive $(\epsilon, \delta)$-LDP, in the rest of the paper we will focus on non-interactive $\epsilon$-LDP algorithms.

[3]Note that here we use an weak version of their result

[4]Note that since here we assume $n$ is at least exponential in $p$, thus the algorithm is not fully polynomial.

[5]A convex set is isotropic if a random vector chosen uniformly from $\mathcal{K}$ according to the volume is isotropic. A random vector $a$ is isotropic if for all $b \in \mathbb{R}^p, \mathbb{E}[\langle a, b\rangle^2] = \|b\|^2$, such as polytope.

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
