[Supplementary Material]

# Supplementary Material of Empirical Risk Minimization in Non-interactive Local Differential Privacy Revisited

**Di Wang**     **Marco Gaboardi**     **Jinhui Xu** [*]
Department of Computer Science and Engineering
State University of New York at Buffalo
Buffalo, NY, 14260

## A   LDP Algorithms for Learning K-way Marginals Queries and Smooth Queries By using Polynomial Approximation

In this section, we will show further applications of our idea by giving $\epsilon$-LDP algorithms for answering sets of queries. All the queries we consider in this section are linear, that is, of the form $q_f(D) = \frac{1}{|D|} \sum_{x \in D} f(x)$ for some function $f$. It will be convenient to have a notion of accuracy for the algorithm we will present with respect to a set of queries. This is defined as follow:

**Definition 1.** Let $\mathcal{Q}$ denote a set of queries. An algorithm $\mathcal{A}$ is said to have $(\alpha, \beta)$-accuracy for size $n$ databases with respect to $\mathcal{Q}$, if for every $n$-size dataset $D$, the following holds: $\Pr[\exists q \in \mathcal{Q}, |\mathcal{A}(D, q) - q(D)| \geq \alpha] \leq \beta$.

### A.1   K-way Marginals Queries

Now we consider a database $D = (\{0,1\}^p)^n$, where each row corresponds to an individuals record. A marginal query is specified by a set $S \subseteq [p]$ and a pattern $t \in \{0,1\}^{|S|}$. Each such query asks: 'What fraction of the individuals in $D$ has each of the attributes set to $t_j$?'. We will consider here k-way marginals which are the subset of marginal queries specified by a set $S \subseteq [p]$ with $|S| \leq k$. K-way marginals permit to represent several statistics over datasets, including contingency tables, and the problem to release them under differential privacy has been studied extensively in the literature [7, 6, 13, 5]. All these previous works have considered the central model of differential privacy, and only the recent work [9] studies this problem in the local model, while their methods are based Fourier Transform. We now use the LDP version of Chebyshev polynomial approximation to give an efficient way of constructing a sanitizer for releasing k-way marginals.

Since learning the class of $k$-way marginals is equivalent to learning the class of monotone k-way disjunctions [7], we will only focus on the latter. The reason why we can locally privately learning them is that they form a $\mathcal{Q}$-Function Family.

**Definition 2** ($\mathcal{Q}$-Function Family). Let $\mathcal{Q} = \{q_y\}_{y \in Y_{\mathcal{Q}} \subseteq \{0,1\}^m}$ be a set of counting queries on a data universe $\mathcal{D}$, where each query is indexed by an $m$-bit string. We define the index set of $\mathcal{Q}$ to be the set $Y_{\mathcal{Q}} = \{y \in \{0,1\}^m | q_y \in \mathcal{Q}\}$.
We define a $\mathcal{Q}$-Function Family $\mathcal{F}_{\mathcal{Q}} = \{f_{\mathcal{Q},x} : \{0,1\}^m \mapsto \{0,1\}\}_{x \in \mathcal{D}}$ as follows: for every data record $x \in D$, the function $f_{\mathcal{Q},x} : \{0,1\}^m \mapsto \{0,1\}$ is defined as $f_{\mathcal{Q},x}(y) = q_y(x)$. Given a database $D \in \mathcal{D}^n$, we define $f_{\mathcal{Q},D}(y) = \frac{1}{n} \sum_{i=1}^{n} f_{\mathcal{Q},x^i}(y) = \frac{1}{n} \sum_{i=1}^{n} q_y(x^i) = q_y(D)$, where $x^i$ is the $i$-th row of $D$.

---

[*]This research was supported in part by the National Science Foundation (NSF) under Grant No. CCF-1422324, CCF-1716400, CCF-1718220 and CNS-1565365.

This definition guarantees that $\mathcal{Q}$-function queries can be computed from their values on the individual's data $x^i$. We can now formally define the class of monotone k-way disjunctions.

**Definition 3.** Let $\mathcal{D} = \{0,1\}^p$. The query set $\mathcal{Q}_{disj,k} = \{q_y\}_{y \in Y_k \subseteq \{0,1\}^p}$ of monotone $k$-way disjunctions over $\{0,1\}^p$ contains a query $q_y$ for every $y \in Y_k = \{y \in \{0,1\}^p | \|y\| \leq k\}$. Each query is defined as $q_y(x) = \vee_{j=1}^p y_j x_j$. The $\mathcal{Q}_{disj,k}$-function family $\mathcal{F}_{\mathcal{Q}_{disj,k}} = \{f_x\}_{x \in \{0,1\}^p}$ contains a function $f_x(y_1, y_2, \cdots, y_p) = \vee_{j=1}^p y_j x_j$ for each $x \in \{0,1\}^p$.

Definition 2 guarantees that if we can uniformly approximated the function $f_{\mathcal{Q},x}$ by polynomials $p_x$, then we can also have an approximation of $f_{\mathcal{Q},D}$, *i.e.* we can approximate $q_y(D)$ for every $y$ or all the queries in the class $\mathcal{Q}$. Thus, if we can locally privately estimate the sum of coefficients of the monomials for the $m$-multivariate functions $\{p_x\}_{x \in D}$, we can uniformly approximate $f_{\mathcal{Q},D}$. Clearly, this can be done by Lemma 2, if the coefficients of the approximated polynomial are bounded.

In order to uniformly approximate the class $\mathcal{Q}_{disj,k}$, we use Chebyshev polynomials.

**Definition 4** (Chebyshev Polynomials). For every $k \in \mathbb{N}$ and $\gamma > 0$, there exists a univariate real polynomial $p_k(x) = \sum_{j=0}^{t_k} c_i x^i$ of degree $t_k$ such that $t_k = O(\sqrt{k} \log(\frac{1}{\gamma}))$; for every $i \in [t_k], |c_i| \leq 2^{O(\sqrt{k} \log(\frac{1}{\gamma}))}$; and $p(0) = 0, |p_k(x) - 1| \leq \gamma, \forall x \in [k]$.

---

**Algorithm 1** Local Chebyshev Mechanism for $\mathcal{Q}_{disj,k}$

1: **Input:** Player $i \in [n]$ holding data $x_i \in \{0,1\}^p$, privacy parameter $\epsilon > 0$, error bound $\alpha$, and $k \in \mathbb{N}$.
2: **for** Each Player $i \in [n]$ **do**
3:     Consider the $p$-multivariate polynomial $q_{x_i}(y_1, \ldots, y_p) = p_k(\sum_{j=1}^p y_j[x_i]_j)$, where $p_k$ is defined as in Definition 4 with $\gamma = \frac{\alpha}{2}$.
4:     Denote the coefficients of $q_{x_i}$ as a vector $\tilde{q}_i \in \mathbb{R}^{\binom{p+t_k}{t_k}}$(since there are $\binom{p+t_k}{t_k}$ coefficients in a $p$-variate polynomial with degree $t_k$), note that each $\tilde{q}_i$ can bee seen as a $p$-multivariate polynomial $q_{x_i}(y)$.
5: **end for**
6: **for** The Server **do**
7:     Run LDP-AVG from Lemma 1 on $\{\tilde{q}_i\}_{i=1}^n \in \mathbb{R}^{\binom{p+t_k}{t_k}}$ with parameter $\epsilon, b = p^{O(\sqrt{k} \log(\frac{1}{\gamma}))}$, denote the output as $\tilde{p}_D \in \mathbb{R}^{\binom{p+t_k}{t_k}}$, note that $\tilde{p}_D$ also corresponds to a $p$-multivariate polynomial.
8:     For each query $y$ in $\mathcal{Q}_{disj,k}$ (seen as a $d$ dimension vector), compute the $p$-multivariate polynomial $\tilde{p}_D(y_1, \ldots, y_p)$.
9: **end for**

---

**Lemma 1.** [13] For every $k, p \in \mathbb{N}$, such that $k \leq p$, and every $\gamma > 0$, there is a family of $p$-multivariate polynomials of degree $t = O(\sqrt{k} \log(\frac{1}{\gamma}))$ with coefficients bounded by $T = p^{O(\sqrt{k} \log(\frac{1}{\gamma}))}$, which uniformly approximate the family $\mathcal{F}_{\mathcal{Q}_{disj,k}}$ over the set $Y_k$ (Definition 3) with error bound $\gamma$. That is, there is a family of polynomials $\mathcal{P}$ such that for every $f_x \in \mathcal{F}_{\mathcal{Q}_{disj,k}}$, there is $p_x \in \mathcal{P}$ which satisfies $\sup_{y \in Y_k} |p_x(y) - f_x(y)| \leq \gamma$.

By combining the ideas discussed above and Lemma 1, we have Algorithm 1 and the following theorem.

**Theorem 1.** For $\epsilon > 0$ Algorithm 1 is $\epsilon$-LDP. Also, for $0 < \beta < 1$, there are constants $C, C_1$ such that for every $k, p, n \in \mathbb{N}$ with $k \leq p$, if $n \geq \Omega(\max\{\frac{p^{C\sqrt{k}\log\frac{1}{\alpha}} \log\frac{1}{\beta}}{\epsilon^2 \alpha^2}, \frac{\log\frac{1}{\beta}}{\epsilon^2}, p^{C_1\sqrt{k}\log\frac{1}{\alpha}} \log\frac{1}{\beta}\})$, this algorithm is $(\alpha, \beta)$-accuracy with respect to $\mathcal{Q}_{disj,k}$. The running time for player is $\text{Poly}(p^{O(\sqrt{k}\log\frac{1}{\alpha})})$, and the running time for server is at most $O(n)$ and the time for answering a query is $O(p^{C_2\sqrt{k}\log\frac{1}{\alpha}})$ for some constant $C_2$. Moreover, as in Section 5, the communication complexity can be improved to 1-bit per player.

*Proof.* It is sufficient to prove that

$$\sup_{y \in Y_k} |\tilde{p}_D(y) - q_y(D)| \leq \gamma + \frac{T\binom{p+t_k}{t_k}^2 \sqrt{\log \frac{\binom{p+t_k}{t_k}}{\beta}}}{\sqrt{n}\epsilon},$$

where $T = p^{O(\sqrt{k}\log(\frac{1}{\gamma}))}$. Now we denote $p_D \in \mathbb{R}^{\binom{p+t_k}{t_k}}$ as the average of $\tilde{q}_i$. That is, it is the unperturbed version of $\tilde{p}_D$. By Lemma 4, we have $\sup_{y \in Y_k} |p_D(y) - q_y(D)| \leq \gamma$. Thus it is sufficient to prove that

$$\sup_{y \in Y_k} |\tilde{p}_D(y) - p_D(y)| \leq \frac{T\binom{p+t_k}{t_k}^2 \sqrt{\log \frac{\binom{p+t_k}{t_k}}{\beta}}}{\sqrt{n}\epsilon}.$$

Since $\tilde{p}_D, p_D$ can be viewed as a vector, we have

$$\sup_{y \in Y_k} |\tilde{p}_D(y) - p_D(y)| \leq \|\tilde{p}_D - p_D\|_1.$$

Also, since each coordinate of $p_D(y)$ is bounded by $T$ by Lemma 1, we can see that if $n = \Omega(\max\{\frac{1}{\epsilon^2} \log \frac{1}{\beta}, \binom{p+t_k}{t_k} \log \binom{p+t_k}{t_k} \log 1/\beta\})$, then with probability at least $1 - \beta$, the following is

true $\|\tilde{p}_D - p_D\|_1 \leq \frac{T\binom{p+t_k}{t_k}^2 \sqrt{\log \frac{\binom{p+t_k}{t_k}}{\beta}}}{\sqrt{n}\epsilon}$, thus take $\gamma = \frac{\alpha}{2}$ and $\binom{p+t_k}{t_k} = p^{O(t_k)}$. This gives us the theorem. $\qquad\square$

## A.2 Smooth Queries

We now consider the case where each player $i \in [n]$ holds a data $x_i \in \mathbb{R}^p$ and we want to estimate the kernel density for a given point $x_0 \in \mathbb{R}^p$. A natural question is: If we want to estimate Gaussian kernel density of a given point $x_0$ with many different bandwidths, can we do it simultaneously under $\epsilon$ local differential privacy?

We can see this kind of queries as a subclass of the smooth queries. So, like in the case of k-way marginals queries, we will give an $\epsilon$-LDP sanitizer for smooth queries. Now we consider the data universe $\mathcal{D} = [-1,1]^p$, and dataset $D \in \mathcal{D}^n$. For a positive integer $h$ and constant $T > 0$, we denote the set of all $p$-dimensional $(h,T)$-smooth function (Definition **??**) as $C_T^h$, and $\mathcal{Q}_{C_T^h} = \{q_f(D) = \frac{1}{n}\sum_{x \in D} f(D), f \in C_T^h\}$ the corresponding set of queries. The idea of the algorithm is similar to the one used for the k-way marginals; but instead of using Chebyshev polynomials, we will use trigonometric polynomials. We now assume that the dimensionality $p$, $h$ and $T$ are constants so all the result in big $O$ notation will be omitted. The idea of Algorithm **??** is actually based on the following Lemma.

**Lemma 2.** [16] Assume $\gamma > 0$. For every $f \in C_T^h$, defined on $[-1,1]^p$, let $g_f(\theta_1, \ldots, \theta_p) = f(\cos(\theta_1), \ldots, \cos(\theta_p))$, for $\theta_i \in [-\pi, \pi]$. Then there is an even trigonometric polynomial $p$ whose degree for each variable is $t(\gamma) = (\frac{1}{\gamma})^{\frac{1}{h}}$:

$$p(\theta_1, \ldots, \theta_p) = \sum_{0 \leq r_1, \ldots, r_p < t(\gamma)} c_{r_1, \ldots, r_p} \prod_{i=1}^{p} \cos(r_i \theta_i), \qquad (1)$$

such that 1) $p$ $\gamma$-uniformly approximates $g_f$, i.e. $\sup_{x \in [-\pi,\pi]^p} |p(x) - g_f(x)| \leq \gamma$. 2) The coefficients are uniformly bounded by a constant $M$ which only depends on $h, T$ and $p$. 3) Moreover, the whole set of the coefficients can be computed in time $O\big((\frac{1}{\gamma})^{\frac{p+2}{h} + \frac{2p}{h^2}} \text{poly} \log \frac{1}{\gamma}\big)$.

By (1), we can see that all the $p(x)$ which corresponds to $g_f(x)$, representing functions $f \in C_T^h$, have the same basis $\prod_{i=1}^{p} \cos(r_i \theta_i)$. So, we can use Lemma 1 or 2 to estimate the average of the basis. Then, for each query $f$ the server can only compute the corresponding coefficients $\{c_{r_1, r_2, \cdots, r_p}\}$. This idea is implemented in Algorithm 2 for which we have the following result.

**Theorem 2.** For $\epsilon > 0$, Algorithm 2 is $\epsilon$-LDP. Also for $\alpha > 0$, $0 < \beta < 1$, if $n \geq \Omega(\max\{\log^{\frac{5p+2h}{2h}}(\frac{1}{\beta})\epsilon^{-2}\alpha^{-\frac{5p+2h}{h}}, \frac{1}{\epsilon^2}\log(\frac{1}{\beta})\})$ and $t = O((\sqrt{n}\epsilon)^{\frac{2}{5p+2h}})$, then Algorithm 2 is $(\alpha, \beta)$-accurate with respect to $\mathcal{Q}_{C_T^h}$. The time for answering each query is $\tilde{O}((\sqrt{n}\epsilon)^{\frac{4p+4}{5p+2h} + \frac{4p}{5ph+2h^2}})$, where $O$ omits $h, T, p$ and some log terms. For each player, the computation and communication cost could be improved to $O(1)$ and 1 bit, respectively, as in Section 5.

*Proof of Theorem 9.* Let $t = (\frac{1}{\gamma})^{\frac{1}{h}}$. It is sufficient to prove that $\sup_{q_f \in \mathcal{Q}_{C_T^h}} |\tilde{p}_D \cdot c_f - q_f(D)| \leq \alpha$.

Let $p_D$ denote the average of $\{p_i\}_{i=1}^n$, *i.e.* the unperturbed version of $\tilde{p}_D$. Then by Lemma 5, we have

---

**Algorithm 2** Local Trigonometry Mechanism for $\mathcal{Q}_{C_T^h}$

---

1: **Input:** Player $i \in [n]$ holding data $x_i \in [-1, 1]^p$, privacy parameter $\epsilon > 0$, error bound $\alpha$, and $t \in \mathbb{N}$. $\mathcal{T}_t^p = \{0, 1, \cdots, t-1\}^p$. For a vector $x = (x_1, \ldots, x_p) \in [-1, 1]^p$, denote operators $\theta_i(x) = \arccos(x_i), i \in [p]$.
2: **for** Each Player $i \in [n]$ **do**
3:     **for** Each $v = (v_1, v_2, \cdots, v_p) \in \mathcal{T}_t^p$ **do**
4:         Compute $p_{i;v} = \cos(v_1\theta_1(x_i)) \cdots \cos(v_p\theta_p(x_i))$
5:     **end for**
6:     Let $p_i = (p_{i;v})_{v \in \mathcal{T}_t^p}$.
7: **end for**
8: **for** The Server **do**
9:     Run LDP-AVG from Lemma 1 on $\{p_i\}_{i=1}^n \in \mathbb{R}^{t^p}$ with parameter $\epsilon$, $b = 1$, denote the output as $\tilde{p}_D$.
10:     For each query $q_f \in \mathcal{Q}_{C_T^h}$. Let $g_f(\theta) = f(\cos(\theta_1), \cos(\theta_2), \cdots, \cos(\theta_p))$.
11:     Compute the trigonometric polynomial approximation $p_t(\theta)$ of $g_f(\theta)$, where $p_t(\theta) = \sum_{r=(r_1, r_2 \cdots r_p), \|r\|_\infty \le t-1} c_r \cos(r_1\theta_1) \cdots \cos(r_p\theta_p)$ as in (1). Denote the vector of the coefficients $c \in \mathbb{R}^{t^p}$.
12:     Compute $\tilde{p}_D \cdot c$.
13: **end for**

---

$\sup_{q_f \in \mathcal{Q}_{C_T^h}} |p_D \cdot c_f - q_f(D)| \le \gamma$. Also since $\|c_f\|_\infty \le M$, we have $\sup_{q_f \in \mathcal{Q}_{C_T^h}} |\tilde{p}_D \cdot c_f - p_D \cdot c_f| \le O(\|\tilde{p}_D - p_D\|_1)$. By Lemma 2, we know that if $n = \Omega(\max\{\frac{1}{\epsilon^2}\log\frac{1}{\beta}, t^{2p}\log\frac{1}{\beta}\})$, then $\|\tilde{p}_D - p_D\|_1 \le O(\frac{t^{\frac{5p}{2}}\sqrt{\log(\frac{1}{\beta})}}{\sqrt{n}\epsilon})$ with probability at least $1-\beta$. Thus, we have $\sup_{q_f \in \mathcal{Q}_{C_T^h}} |\tilde{p}_D \cdot c_f - q_f(D)| \le O(\gamma + \frac{(\frac{1}{\gamma})^{\frac{5p}{2h}}\sqrt{\log(\frac{1}{\beta})}}{\sqrt{n}\epsilon})$. Taking $\gamma = O((1/\sqrt{n}\epsilon)^{\frac{2h}{5p+2h}})$, we get $\sup_{q_f \in \mathcal{Q}_{C_T^h}} |\tilde{p}_D \cdot c_f - q_f(D)| \le O(\sqrt{\log(\frac{1}{\beta})}(\frac{1}{\sqrt{n}\epsilon})^{\frac{2h}{5p+2h}}) \le \alpha$. The computational cost for answering a query follows from Lemma 2 and $b \cdot c = O(t^p)$. $\qquad\square$

## B   Details in Section 3

**Lemma 3.** [10] Suppose that $x_1, \cdots, x_n$ are i.i.d sampled from $\text{Lap}(\frac{1}{\epsilon})$. Then for every $0 \le t < \frac{2n}{\epsilon}$, we have

$$\Pr(|\sum_{i=1}^n x_i| \ge t) \le 2\exp(-\frac{\epsilon^2 t^2}{4n}).$$

*Proof of Lemma 1.* Consider Algorithm 1. We have $|a - \frac{1}{n}\sum_{i=1}^n v_i| = |\frac{\sum_{i=1}^n x_i}{n}|$, where $x_i \sim \text{Lap}(\frac{b}{\epsilon})$. Taking $t = \frac{2\sqrt{n}\sqrt{\log\frac{2}{\beta}}}{\epsilon}$ and applying the above lemma, we prove the lemma. $\qquad\square$

## C   Details in Section 4

### C.1   Proof of Theorem 3

*Proof of Theorem 3.* The proof of the $\epsilon$-LDP comes from Lemma 1 and composition theorem. W.l.o.g, we assume T=1. To prove the theorem, it is sufficient to estimate $\sup_{\theta \in \mathcal{C}} |\tilde{L}(\theta; D) - \hat{L}(\theta; D)| \le \alpha$ for some $\alpha$, since if it is true, denote $\theta^* = \arg\min_{\theta \in \mathcal{C}} \hat{L}(\theta; D)$, we have $\hat{L}(\theta_{\text{priv}}; D) - \hat{L}(\theta^*; D) \le \hat{L}(\theta_{\text{priv}}; D) - \tilde{L}(\theta_{\text{priv}}; D) + \tilde{L}(\theta_{\text{priv}}; D) - \tilde{L}(\theta^*; D) + \tilde{L}(\theta^*; D) - \hat{L}(\theta^*; D) \le \hat{L}(\theta_{\text{priv}}; D) - \tilde{L}(\theta_{\text{priv}}; D) + \tilde{L}(\theta^*; D) - \hat{L}(\theta^*; D) \le 2\alpha$.

Since we have $\sup_{\theta \in \mathcal{C}} |\tilde{L}(\theta; D) - \hat{L}(\theta; D)| \le \sup_{\theta \in \mathcal{C}} |\tilde{L}(\theta; D) - B_k^{(h)}(\hat{L}, \theta)| + \sup_{\theta \in \mathcal{C}} |B_k^{(h)}(\hat{L}, \theta) - \hat{L}(\theta; D)|$. The second term is bounded by $O(D_h p\frac{1}{k^h})$ by Theorem 2.

For the First term, by (2) and the algorithm, we have

$$\sup_{\theta \in \mathcal{C}} |\tilde{L}(\theta; D) - B_k^{(h)}(\hat{L}, \theta)| \leq \max_{v \in \mathcal{T}} |\tilde{L}(v; D) - \hat{L}(v; D)| \sup_{\theta \in \mathcal{C}} \sum_{j=1}^{p} \sum_{v_j=0}^{k} |\prod_{i=1}^{p} b_{v_i,k}^{(h)}(\theta_i)|. \quad (2)$$

By Proposition 4 in [1], we have $\sum_{j=1}^{p} \sum_{v_j=0}^{k} |\prod_{i=1}^{p} b_{v_i,k}^{(h)}(\theta_i)| \leq (2^h - 1)^p$. Next lemma bounds the term $\max_{v \in \mathcal{T}} |\tilde{L}(v; D) - \hat{L}(v; D)|$, which is obtained by Lemma 1.

**Lemma 4.** If $0 < \beta < 1$, $k$ and $n$ satisfy that $n \geq p \log(2/\beta) \log(k+1)$, then with probability at least $1 - \beta$, for each $v \in \mathcal{T}$,

$$|\tilde{L}(v; D) - \hat{L}(v; D)| \leq O\left(\frac{\sqrt{\log \frac{1}{\beta}} \sqrt{p} \sqrt{\log(k)}(k+1)^p}{\sqrt{n}\epsilon}\right). \quad (3)$$

*Proof.* By Lemma 1, for a fixed $v \in \mathcal{T}$, if $n \geq \log \frac{2}{\beta}$, we have with probability $1 - \beta$, $|\tilde{L}(v; D) - \hat{L}(v; D)| \leq \frac{2\sqrt{\log \frac{2}{\beta}}}{\sqrt{n}\epsilon}$. Taking the union of all $v \in \mathcal{T}$ and then taking $\beta = \frac{\beta}{(k+1)^p}$ (since there are $(k+1)^p$ elements in $\mathcal{T}$) and $\epsilon = \frac{\epsilon}{(k+1)^p}$, we get the proof. $\square$

By $(k+1) < 2k$, we have

$$\sup_{\theta \in \mathcal{C}} |\tilde{L}(\theta; D) - \hat{L}(\theta; D)| \leq O\left(\frac{D_h p}{k^h} + \frac{2^{(h+1)p}\sqrt{\log \frac{1}{\beta}}\sqrt{p \log k} k^p}{\sqrt{n}\epsilon}\right). \quad (4)$$

Now we take $k = O\left(\frac{D_h \sqrt{pn}\epsilon}{2^{(h+1)p}\sqrt{\log \frac{1}{\beta}}}\right)^{\frac{1}{h+p}}$. Since $n = \Omega\left(\frac{4^{p(h+1)}}{\epsilon^2 p D_h^2}\right)$, we have $\log k > 1$. Pluggning it into (4), we get

$$\sup_{\theta \in \mathcal{C}} |\tilde{L}(\theta; D) - \hat{L}(\theta; D)| \leq \tilde{O}\left(\frac{\log^{\frac{h}{2(h+p)}}(\frac{1}{\beta}) D_h^{\frac{p}{p+h}} p^{\frac{1}{2}+\frac{p}{2(h+p)}} 2^{(h+1)p \frac{h}{h+p}}}{\sqrt{h+p} n^{\frac{h}{2(h+p)}} \epsilon^{\frac{h}{h+p}}}\right) = \tilde{O}\left(\frac{\log^{\frac{h}{2(h+p)}}(\frac{1}{\beta}) D_h^{\frac{p}{p+h}} p^{\frac{p}{2(h+p)}} 2^{(h+1)p}}{n^{\frac{h}{2(h+p)}} \epsilon^{\frac{h}{h+p}}}\right). \quad (5)$$

Also we can see that $n \geq p \log(2/\beta) \log(k+1)$ is true for $n = \Omega\left(\frac{4^{p(h+1)}}{\epsilon^2 p D_h^2}\right)$. Thus, the theorem follows. $\square$

*Proof of Corollary 1 and 2.* Since the loss function is $(\infty, T)$-smooth, it is $(2p, T)$-smooth for all $p$. Thus, taking $h = p$ in Theorem 3, we get the proof. $\square$

### C.2 Population Risk of Algorithm 2

Here we will only show the case of $(\infty, T)$, it is the same for the general case.

**Theorem 3.** Under the conditions in Corollary 2, if we further assume the loss function $\ell(\cdot, x)$ to be convex and 1-Lipschitz for all $x \in \mathcal{D}$ and the convex set $\mathcal{C}$ satisfying $\|\mathcal{C}\|_2 \leq 1$, then with probability at least $1 - 2\beta$, we have: $\text{Err}_{\mathcal{P}}(\theta_{\text{priv}}) \leq \tilde{O}\left(\frac{(\sqrt{\log 1/\beta})^{\frac{1}{4}} D_p^{\frac{1}{4}} p^{\frac{1}{8}} c_1^{p^2}}{\beta n^{\frac{1}{12}} \epsilon^{\frac{1}{4}}}\right)$. That is, if we have sample complexity $n = \tilde{\Omega}\left(\max\{\frac{\log \frac{1}{\beta} c^{p^2}}{\epsilon^2 D_p^2}, (\sqrt{\log 1/\beta})^3 D_p^3 p^{\frac{3}{2}} c_2^{p^2} \epsilon^{-3} \alpha^{-12} \beta^{-12}\}\right)$, then we have $\text{Err}_{\mathcal{P}}(\theta_{\text{priv}}) \leq \alpha$. Here $c, c_1, c_2$ are some constants.

**Lemma 5.** [11] If the loss function $\ell$ is L-Lipschitz and $\mu$-strongly convex, then with probability at least $1 - \beta$ over the randomness of sampling the data set $\mathcal{D}$, the following is true,

$$\text{Err}_{\mathcal{P}}(\theta) \leq \sqrt{\frac{2L^2}{\mu}} \sqrt{\text{Err}_{\mathcal{D}}(\theta)} + \frac{4L^2}{\beta \mu n}. \quad (6)$$

*Proof of Theorem 3.* For the general convex loss function $\ell$, we let $\hat{\ell}(\theta; x) = \ell(\theta; x) + \frac{\mu}{2}\|\theta\|^2$ for some $\mu > 0$. Note that in this case the new empirical risk becomes $\bar{L}(\theta; D) = \hat{L}(\theta; D) + \frac{\mu}{2}\|\theta\|^2$. Since $\frac{\mu}{2}\|\theta\|^2$ does not depend on the dataset, we can still use the Bernstein polynomial approximation for the original empirical risk $\hat{L}(\theta; D)$ as in Algorithm 2, and the error bound for $\bar{L}(\theta; D)$ is the same. Thus, we can get the population excess risk of the loss function $\hat{\ell}$, $\text{Err}_{\mathcal{P}, \hat{\ell}}(\theta_{\text{priv}})$ by Corollary 1 and we have the following relation,

$$\text{Err}_{\mathcal{P}, \ell}(\theta_{\text{priv}}) \leq \text{Err}_{\mathcal{P}, \hat{\ell}}(\theta_{\text{priv}}) + \frac{\mu}{2}.$$

By the above lemma for $\text{Err}_{\mathcal{P}, \hat{\ell}}(\theta_{\text{priv}})$, where $\hat{\ell}(\theta; x)$ is $1 + \|\mathcal{C}\|_2 = O(1)$-Lipschitz, thus we have the following,

$$\text{Err}_{\mathcal{P}, \ell}(\theta_{\text{priv}}) \leq \tilde{O}\left(\sqrt{\frac{2}{\mu}} \frac{\log^{\frac{1}{8}} \frac{1}{\beta} D_p^{\frac{1}{4}} p^{\frac{1}{8}} c^{(p+1)p}}{n^{\frac{1}{8}} \epsilon^{\frac{1}{4}}} + \frac{4}{\beta\mu n} + \frac{\mu}{2}\right).$$

Taking $\mu = O(\frac{1}{\sqrt[12]{n}})$, we get

$$\text{Err}_{\mathcal{P}, \ell}(\theta_{\text{priv}}) \leq \tilde{O}\left(\frac{\log^{\frac{1}{8}} \frac{1}{\beta} D_p^{\frac{1}{4}} p^{\frac{1}{8}} c^{p^2}}{\beta n^{\frac{1}{12}} \epsilon^{\frac{1}{4}}}\right).$$

Thus, we have the theorem. $\qquad\qquad\qquad\qquad\qquad\qquad\qquad\qquad\qquad\qquad\qquad\qquad\quad\square$

## D  Details in Section 5

---

**Algorithm 3** Player-Efficient Local Bernstein Mechanism with $O(\log n)$-bits communication per player

---

1: **Input:** Each user $i \in [n]$ has data $x_i \in \mathcal{D}$, privacy parameter $\epsilon$, public loss function $\ell : [0,1]^p \times \mathcal{D} \mapsto [0,1]$, and parameter $k$( we will specify it later).
2: **Preprocessing:**
3:  Construct the grid $\mathcal{T} = \{\frac{v_1}{k}, \frac{v_2}{k}, \cdots, \frac{v_p}{k}\}_{v_1, v_2, \cdots, v_p}$, where $\{v_1, v_2, \cdots, v_p\} = \{0, 1, \cdots, k\}^p$.
4:  Discretize the interval $[0,1]$ with grid steps $O(\frac{1}{n\epsilon}\sqrt{\frac{d}{n}\log(\frac{d}{\beta})})$. Denote the set of grids by $\mathcal{G}$.
5:  Randomly partition $[n]$ in to $d = (k+1)^p$ subsets $I_1, I_2, \cdots, I_d$, with each subset $I_j$ corresponding to a grid in $\mathcal{T}$ denoted as $\mathcal{T}(j)$.
6: **for** Each Player $i \in [n]$ **do**
7:      Find the subset $I_\ell$ such that $i \in I_\ell$. Calculate $v_i = \ell(\mathcal{T}(l); x_i)$.
8:      Denote $z_i = v_i + \text{Lap}(\frac{1}{\epsilon})$, round $z_i$ into the grid set $\mathcal{G}$, and let the resulting one be $\tilde{z}_i$.
9:      Send $(\tilde{z}_i, \ell)$.
10: **end for**
11: **for** The Server **do**
12:     **for** Each $\ell \in [d]$ **do**
13:         Compute $v_\ell = \frac{n}{|I_\ell|} \sum_{i \in I_\ell} \tilde{z}_i$.
14:         Denote the corresponding grid point $(\frac{v_1}{k}, \frac{v_2}{k}, \cdots, \frac{v_p}{k}) \in \mathcal{T}$ as $\ell$; then let $\hat{L}((\frac{v_1}{k}, \frac{v_2}{k}, \cdots, \frac{v_p}{k}); D) = v_\ell$.
15:     **end for**
16:     Construct perturbed Bernstein polynomial of the empirical loss $\tilde{L}$ as in Algorithm 2, where each $\hat{L}((\frac{v_1}{k}, \frac{v_2}{k}, \cdots, \frac{v_p}{k}); D)$ is replaced by $\tilde{L}((\frac{v_1}{k}, \frac{v_2}{k}, \cdots, \frac{v_p}{k}); D)$. Denote the function as $\tilde{L}(\cdot, D)$.
17:     Compute $\theta_{\text{priv}} = \arg\min_{\theta \in \mathcal{C}} \tilde{L}(\theta; D)$.
18: **end for**

---

*Proof of Theorem 4.* By [2] it is $\epsilon$-LDP. The time complexity and communication complexity is obvious. As in [2], it is sufficient to show that the LDP-AVG is sampling resilient. Here the STAT is the average, and $\phi(x, y)$ is $\max_{j \in [p]} |[x]_j - [y]_j|$. By Lemma 2, we can see that with probability at least $1 - \beta$, $\phi(\text{Avg}(v_1, v_2, \cdots, v_n); a) = O(\frac{bp}{\sqrt{n}\epsilon}\sqrt{\log\frac{p}{\beta}})$. Now let $\mathcal{S}$ be the set obtained by sampling

each point $v_i, i \in [n]$ independently with probability $\frac{1}{2}$. Note that by Lemma 2, we have on the subset $\mathcal{S}$. If $|S| \geq \Omega(\max\{p\log(\frac{p}{\beta}), \frac{1}{\epsilon^2}\log\frac{1}{\beta}\})$ with probability $1 - \beta$, $\phi(\text{Avg}(\mathcal{S}); \text{LDP-AVG}(\mathcal{S})) = O(\frac{b\sqrt{p}}{\sqrt{|\mathcal{S}|}\epsilon}\sqrt{\log\frac{p}{\beta}})$. Now by Hoeffdings Inequality, we can get $|n/2 - |\mathcal{S}|| \leq \sqrt{n\log\frac{4}{\beta}}$ with probability $1 - \beta$. Also since $n = \Omega(\log\frac{1}{\beta})$, we know that $|\mathcal{S}| \geq O(n) \geq \Omega(p\log(\frac{p}{\beta}))$ is true. Thus, with probability at least $1 - 2\beta$, $\phi(\text{Avg}(\mathcal{S}); \text{LDP-AVG}(\mathcal{S})) = O(\frac{bp}{\sqrt{n}\epsilon}\sqrt{\log\frac{p}{\beta}})$.

Actually, we can also get $\phi(\text{Avg}(\mathcal{S}); \text{Avg}(v_1, v_2, \cdots, v_n)) \leq O(\frac{bd}{\sqrt{n}\epsilon}\sqrt{\log\frac{d}{\beta}})$. We now first assume that $v_i \in \mathbb{R}$. Note that $\text{Avg}(\mathcal{S}) = \frac{v_1 x_1 + \cdots + v_n x_n}{x_1 + \cdots + x_n}$, where each $x_i \sim \text{Bernoulli}(\frac{1}{2})$. Denote $M = x_1 + x_2 + \cdots + x_n$, by Hoeffdings Inequality, we have with probability at least $1 - \frac{\beta}{2}$, $|M - \frac{n}{2}| \leq \sqrt{n\log\frac{4}{\beta}}$. Denote $N = v_1 x_1 + \cdots + v_n x_n$. Also, by Hoeffdings inequality, with probability at least $1 - \beta$, we get $|N - \frac{v_1 + \cdots + v_n}{2}| \leq b\sqrt{n\log\frac{2}{\beta}}$. Thus, with probability at least $1 - \beta$, we have:

$$|\frac{N}{M} - \frac{v_1 + \cdots + v_n}{n}| \leq \frac{|N - \sum_{i=1}^{n} v_i/2|}{M} + |\sum_{i=1}^{n} v_i/2||\frac{1}{M} - \frac{2}{n}| \leq \frac{|N - \sum_{i=1}^{n} v_i/2|}{M} + \frac{nb}{2}|\frac{1}{M} - \frac{2}{n}|.$$
(7)

The second term $|\frac{1}{M} - \frac{2}{n}| = \frac{|n/2 - M|}{M\frac{n}{2}}$. We know from the above $|n/2 - M| \leq \sqrt{n\log\frac{4}{\beta}}$. Also since $n = \Omega(\log\frac{1}{\beta})$, we get $M \geq O(n)$. Thus, $|\frac{1}{M} - \frac{2}{n}| \leq O(\frac{\sqrt{\log\frac{1}{\beta}}}{\sqrt{n}n})$. The upper bound of the second term is $O(\frac{b\sqrt{\log\frac{1}{\beta}}}{\sqrt{n}})$. The same for the first term. For $p$ dimensions, we just choose $\beta = \frac{\beta}{p}$ and take the union. Thus, we have $\phi(\text{Avg}(\mathcal{S}); \text{Avg}(v_1, v_2, \cdots, v_n)) \leq O(\frac{b}{\sqrt{n}\epsilon}\sqrt{\log\frac{p}{\beta}}) \leq O(\frac{bp}{\sqrt{n}\epsilon}\sqrt{\log\frac{p}{\beta}})$.

In summary, we have shown that $\phi(\text{AVG-LDP}(\mathcal{S}); \text{Avg}(v_1, v_2, \cdots, v_n)) \leq O(\frac{bp}{\sqrt{n}\epsilon}\sqrt{\log\frac{p}{\beta}})$ with probability at least $1 - 4\beta$. $\qquad\square$

Recently, [3] proposed a generic transformation, GenProt, which could transform any $(\epsilon, \delta)$ (so as for $\epsilon$) non-interactive LDP protocol to an $O(\epsilon)$-LDP protocol with the communication complexity for each player being $O(\log\log n)$, which removes the condition of 'sample resilient' in [2]. The detail is in Algorithm 2. The transformation uses $O(n\log\frac{n}{\beta})$ independent public string. The reader is referred to [3] for details. Actually, by Algorithm 2, we can easily get an $O(\epsilon)$-LDP algorithm with the same error bound.

**Theorem 4.** With $\epsilon \leq \frac{1}{4}$, under the condition of Corollary 1, Algorithm 4 is $10\epsilon$-LDP. If $T = O(\log\frac{n}{\beta})$, then with probability at least $1 - 2\beta$, Corollary 1 holds. Moreover, the communication complexity of each layer is $O(\log\log n)$ bits, and the computational complexity for each player is $O(\log\frac{n}{\beta})$.

*Proof of Theorem 5.* Let $\theta^* = \arg\min_{\theta \in \mathcal{C}} \hat{L}(\theta; D)$, $\theta_{\text{priv}} = \arg\min_{\theta \in \mathcal{C}} \tilde{L}(\theta; D)$. Under the assumptions of $\alpha, n, k, \epsilon, \beta$, we know from the proof of Theorem 3 and Corollary 1 that $\sup_{\theta \in \mathcal{C}} |\tilde{L}(\theta; D) - \hat{L}(\theta; D)| \leq \alpha$. Also by setting $\epsilon = 16348p\alpha$ and $\alpha \leq \frac{1}{16348}\frac{\mu}{p\sqrt{p}}$, we can see that the condition in Lemma 3 holds for $\Delta = \alpha$. So there is an algorithm returns

$$\tilde{L}(\tilde{\theta}_{\text{priv}}; D) \leq \min_{\theta \in \mathcal{C}} \tilde{L}(\theta; D) + O(p\alpha).$$

Thus, we have

$$\hat{L}(\tilde{\theta}_{\text{priv}}; D) - \hat{L}(\theta^*; D) \leq \hat{L}(\tilde{\theta}_{\text{priv}}; D) - \tilde{L}(\tilde{\theta}_{\text{priv}}; D) + \tilde{L}(\tilde{\theta}_{\text{priv}}; D) - \hat{L}(\theta^*; D),$$

where

$$\hat{L}(\tilde{\theta}_{\text{priv}}; D) - \tilde{L}(\tilde{\theta}_{\text{priv}}; D) \leq \hat{L}(\tilde{\theta}_{\text{priv}}; D) - \tilde{L}(\tilde{\theta}_{\text{priv}}; D) + \tilde{L}(\tilde{\theta}_{\text{priv}}; D) - \tilde{L}(\theta_{\text{priv}}; D) \leq \alpha + O(p\alpha) = O(p\alpha).$$

Also $\tilde{L}(\theta_{\text{priv}}; D) - \hat{L}(\theta^*; D) \leq \tilde{L}(\theta^*; D) - \hat{L}(\theta^*; D) \leq \alpha$. The theorem follows. The running time is determined by $n$. This is because when we use the algorithm in Lemma 3, we have to use the first order optimization. That is, we have to evaluate some points at $\tilde{L}(\theta; D)$, which will cost at most $O(\text{poly}(n))$ time (note that $\tilde{L}$ is a polynomial with $(k+1)^p \leq n$ coefficients). $\qquad\square$

**Algorithm 4** Player-Efficient Local Bernstein Mechanism with $O(\log \log n)$ bits communication complexity.

---

1: **Input:** Each user $i \in [n]$ has data $x_i \in \mathcal{D}$, privacy parameter $\epsilon$, public loss function $\ell :$ $[0,1]^p \times \mathcal{D} \mapsto [0,1]$, and parameter $k, T$.
2: **Preprocessing:**
3: For every $(i, T) \in [n] \times [T]$, generate independent public string $y_{i,t} = \text{Lap}(\perp)$.
4: Construct the grid $\mathcal{T} = \{\frac{v_1}{k}, \frac{v_2}{k}, \cdots, \frac{v_p}{k}\}_{v_1, v_2, \cdots, v_p}$, where $\{v_1, v_2, \cdots, v_p\} = \{0, 1, \cdots, k\}^p$.
5: Randomly partition $[n]$ in to $d = (k+1)^p$ subsets $I_1, I_2, \cdots, I_d$, with each subset $I_j$ corresponding to an grid in $\mathcal{T}$ denoted as $\mathcal{T}(j)$.
6: **for** Each Player $i \in [n]$ **do**
7:    Find the subset $I_\ell$ such that $i \in I_\ell$. Calculate $v_i = \ell(\mathcal{T}(l); x_i)$.
8:    For each $t \in [T]$, compute $p_{i,t} = \frac{1}{2} \frac{\Pr[v_i + Lap(\frac{1}{\epsilon}) = y_{i,t}]}{\Pr[\text{Lap}(\perp) = y_{i,t}]}$
9:    For every $t \in [T]$, if $p_{i,t} \notin [\frac{e^{-2\epsilon}}{2}, \frac{e^{2\epsilon}}{2}]$, then set $p_{i,t} = \frac{1}{2}$.
10:    For every $t \in [T]$, sample a bit $b_{i,t}$ from Bernoulli$(p_{i,t})$.
11:    Denote $H_i = \{t \in [T] : b_{i,t} = 1\}$
12:    If $H_i = \emptyset$, set $H_i = [T]$
13:    Sample $g_i \in H_i$ uniformly, and send $g_i$ to the server.
14: **end for**
15: **for** The Server **do**
16:    **for** Each $l \in [d]$ **do**
17:       Compute $v_\ell = \frac{n}{|I_\ell|} \sum_{i \in I_\ell} g_i$.
18:       Denote the corresponding grid point $(\frac{v_1}{k}, \frac{v_2}{k}, \cdots, \frac{v_p}{k}) \in \mathcal{T}$ as $\ell$; then let $\hat{L}((\frac{v_1}{k}, \frac{v_2}{k}, \cdots, \frac{v_p}{k}); D) = v_\ell$.
19:    **end for**
20:    Construct perturbed Bernstein polynomial of the empirical loss $\tilde{L}$ as in Algorithm 2. Denote the function as $\tilde{L}(\cdot, D)$.
21:    Compute $\theta_{\text{priv}} = \arg\min_{\theta \in \mathcal{C}} \tilde{L}(\theta; D)$.
22: **end for**

---

# E   Details of Section 6

## E.1   Modified $\epsilon$-LDP Algorithm

Note that we cannot use the $\epsilon$-LDP algorithm (see Figure 5 in [12]) in [12] since it needs $n \geq k$, where $k = O\big(\frac{2^{\frac{p-1}{2}}\sqrt{p}}{\alpha^{p-1}}\big)$, and $\alpha = O\big(\big(\frac{\sqrt{p}}{\epsilon^2 n} \log^3(\epsilon^2 n)\big)^{\frac{1}{p+1}}\big)$. This means that $n \geq O(c^p)$, which is contradictory to our assumption. Instead, we will provide a similar algorithm that does not need this assumption. The idea comes from [3], which shows that in non-interactive local model, every $(\epsilon, \delta)$-LDP protocol can be transformed to an $\epsilon$-LDP algorithm. Thus, our idea is the follows. In Figure 5 of [12], instead of partitioning the dataset into $k$ parts and running the subroutine of Figure 1 in [12], we will run $k$ directions for the whole dataset, by the advanced composition theorem (corollary 3.21 in [4]). If for each direction, we run $(\epsilon_0 = O(\frac{\epsilon}{\sqrt{k \log(1/\delta)}}), 0)$-LDP, then the whole LDP algorithm is $(\epsilon, \delta)$-LDP. After that, we use the protocol in [3] to convert the $(\epsilon, \delta)$-LDP algorithm into $O(\epsilon)$-DP. See Algorithm 5. We have the following theorem for Algorithm 5. The proof is the same as in [12]:

**Theorem 5.** Under the same assumption as in Theorem 10 of [12], Algorithm 5 is $(\epsilon, \delta)$-LDP for any $1 > \epsilon > 0$ and $0 < \delta < 1$. Also, for every $k$, with probability at least $1 - \gamma$, the output satisfies

$$\|\hat{f}^j - L_\mathcal{P}\|_\infty \leq O\big(\frac{\log(\epsilon^2 n / k \log(1/\delta))}{\epsilon} \sqrt{\frac{k \log(1/\delta) \log(\epsilon^2 n / \log(1/\delta))\gamma}{n}}\big). \tag{8}$$

Furthermore, if taking $k = O\big(\frac{2^{(p-1)/2} \log(1/\gamma)}{\alpha^{p-1}} \sqrt{\frac{\pi p}{2}}\big)$, where $\alpha = O\big(\big(\frac{\sqrt{p}}{\epsilon^2 n} \log^3(\epsilon^2 n) \log^2(1/\gamma)\big)^{\frac{1}{p+1}}\big)$ and the big-$O$ notation omits the $\log(1/\delta)$ factors, then $\|\hat{f} - L_\mathcal{P}\|_\infty \leq \tilde{O}(\alpha)$ holds with probability at least $1 - 2\gamma$.

**Algorithm 5** $(\epsilon, \delta)$ protocol LDP Algorithm

---

1: **Input:** Each user $i \in [n]$ has data $x_i \in \mathcal{D}$, privacy parameters $\epsilon, \delta$, public loss function
   $\ell : [0,1]^p \times \mathcal{D} \mapsto [0,1]$ satisfies the assumption in [12], and parameter $k$( we will specify it later).
2: **Preprocessing:**
3: Choose $k$ random directions, $u_1, u_2, \cdots, u_k$ and send to each user.
4: **for** Each user $i \in [n]$ **do**
5:     For each $j \in [k]$, invoke 1D-General (Figure 3 in [12]) with $(x_i, u_j)$ with $\epsilon = \frac{\epsilon}{2\sqrt{2k\log(1/\delta)}}$
       and $\gamma = \gamma/k$, output $\mathcal{T}_{i,j}$. Then send $\mathcal{T}_i = (\mathcal{T}_{i,1}, \cdots, \mathcal{T}_{i,k})$ to the server.
6: **end for**
7: **for** The server **do**
8:     After receiving $\{\mathcal{T}_i\}_{i=1}^n$, do the following steps
9:     For $j \in [k]$, invokes 1-D General (Figure 4 in [12]) with $\{\mathcal{T}_{i,j}\}_{i=1}^n$ to get $\hat{f}^j$.
10:    Compute $\theta_j = \arg\min_{\theta || u_j} \hat{f}^j$ and then compute $\theta_{\text{priv}} = \arg\min_j \hat{f}^j(\theta_j)$, output $\theta_{\text{priv}}$.
11: **end for**

---

Now, we have almost the same upper bound as in Theorem 10 of [12]. After using GenProt in [3], we can have an $10\epsilon$-LDP which has the same error bound as in Theorem 5:

**Theorem 6.** Let $\epsilon \leq \frac{1}{4}$. If set $\delta = O(\frac{\epsilon\gamma}{n\ln(2n/\gamma)})$ in Algorithm 5 as the protocol and run the Genprot algorithm in [3], then it is a $10\epsilon$-LDP algorithm whose output $w_{\text{priv}}$ satisfies the following inequality with probability at least $1 - 3\gamma$

$$\text{Err}_{\mathcal{P}}(\theta_{\text{priv}}) \leq \tilde{O}\big((\frac{\sqrt{p}\log^2(1/\beta)}{\epsilon^2 n})^{\frac{1}{p+1}}\big).$$

## E.2   Proof of Theorem 7

Before presenting the proof, we first review some definitions. We refer the reader to [14, 15] for more details.

**Definition 5.** (Sub-Gaussian random vector) A random variable $a \in \mathbb{R}$ is called sub-Gaussian if there exits a constant $C > 0$ such that $\Pr[|a| > t] \leq 2\exp(\frac{-t^2}{C^2})$ for any $t > 0$. Also we say a random vector $a \in \mathbb{R}^p$ is sub-Gaussian if the one dimensional marginals $\langle a, b \rangle$ are sub-Gaussian random variable for all $b \in \mathbb{R}^p$.

For any sub-Gaussian random variable (vector), we can define the sub-Gaussian norm.

**Definition 6.** The $\psi_2$ norm of a sub-Gaussian random variable $a \in \mathbb{R}$, denoted by $\|a\|_{\psi_2}$, is:

$$\|a\|_{\psi_2} = \inf\{t > 0 : \mathbb{E}[\exp(\frac{|a|^2}{t^2})] \leq 2\}.$$

The $\psi_2$ norm of a sub-Gaussian vector $a \in \mathbb{R}^p$ is:

$$\|a\|_{\psi_2} = \sup_{b \in \mathcal{S}^{p-1}} \|\langle a, b \rangle\|_{\psi_2}.$$

Note that when $a$ is normal random Gaussian vector, then $\|a\|_{\psi_2}$ is bounded by a constant [14].

**Definition 7** (Gaussian Width). Given a closed set $S \subset \mathbb{R}^d$, its Gaussian width is defined as:

$$\mathcal{G}_{\mathcal{C}} = \mathbb{E}_{g \sim \mathcal{N}(0,1)^d}[\sup_{a \in S} \langle a, g \rangle].$$

The Minkowski norm (denoted by $\| \cdot \|_{\mathcal{C}}$) with respect to a centrally symmetric convex set $\mathcal{C} \subseteq \mathbb{R}^p$ is defined as follows. For any vector $v \in \mathbb{R}^p$,

$$\| \cdot \|_{\mathcal{C}} = \min\{r \in \mathbb{R}^+ : v \in r\mathcal{C}\}.$$

The main theorem of dimension reduction is as the following:

**Theorem 7.** Let $\tilde{\Phi} \in \mathbb{R}^{m \times p}$ be an random matrix, whose rows are i.i.d mean-zero, isotropic, sub-Gaussian random variable in $\mathbb{R}^d$ with $\psi = \|\Phi_i\|_{\psi_2}$. Let $\Phi = \frac{1}{\sqrt{m}}\tilde{\Phi}$. let $S$ be a set of points in $R^d$. Then there is a constant $C > 0$ such that for any $0 < \gamma, \beta < 1$.

$$\Pr[\sup_{a \in S} |\|\Phi a\|^2 - \|a\|^2 \leq \gamma\|a\|^2] \leq \beta,$$

provided that $m \geq \frac{C\psi^4}{\gamma^2} \max\{\mathcal{G}_\mathcal{S}, \log(1/\beta)\}^2$.

The proof follows from [8]. Below we rephrase it for self-completeness.

**Lemma 6.** Let $\Phi$ be a random matrix as defined in Theorem 7 with $m = \Theta((\frac{\psi^4}{\gamma^2} \log(n/\beta))$ for $\beta > 0$. Then with probability at least $1 - \beta$, $f(\langle \Phi y_i, \cdot \rangle, z_i)$ is 2-Lipschitz over the domain $\Phi \mathcal{C}$ for each $i \in [n]$.

Now since $\mathcal{C}$ is convex, $\Phi \mathcal{C}$ is also convex. Furthermore, by Theorem 7 we know that if $m = \Theta(\frac{\psi^4}{\gamma} \max\{\mathcal{G}_\mathcal{C}^2, \log \frac{1}{\beta}\})$ for $\gamma < 1$, then $\|\Phi \mathcal{C}\|_2 \leq O(1)$. Thus after compression, the loss function and the constrained set still satisfy the assumption in [12]. By Theorem 13, we have:

**Theorem 8.** With probability at least $1 - \beta$,

$$\frac{1}{n}\sum_{i=1}^{n} f(\langle \Phi y_i, \bar{w}\rangle, z_i) - \min_{w \in \mathcal{C}} \frac{1}{n}\sum_{i=1}^{n} f(\langle \Phi y_i, \Phi w\rangle, z_i) \leq \tilde{O}((\frac{\log^2(1/\beta)\sqrt{m}}{n\epsilon^2})^{\frac{1}{m+1}}). \tag{9}$$

We now have the following by using Lipschitz and Theorem 7:

**Lemma 7.** Let $\Phi$ be the random matrix in Theorem 7 with $m = \Theta((\frac{\psi^4}{\gamma^2} \log(n/\beta))$ for $\beta > 0$. Then for any $\hat{w} \in \mathcal{C}$, with probability at least $1 - \beta$, we have

$$\min_{w \in \mathcal{C}} \frac{1}{n}\sum_{i=1}^{n} f(\langle \Phi y_i, \Phi w\rangle, z_i) \leq \frac{1}{n}\sum_{i=1}^{n} f(\langle y_i, \hat{w}\rangle, z_i) + O(\gamma\|\mathcal{C}\|_2.) \tag{10}$$

Let $\theta \in \mathcal{C}$ be the point satisfying the condition $\Phi\theta = \bar{w}$. We have the following lemma by Theorem 7.

**Lemma 8.** Let $\Phi$ be a random matrix as in Theorem 7 with $m = \Theta((\frac{\psi^4}{\gamma^2}(\mathcal{G}_\mathcal{C} + \sqrt{\log n})^2 \log(n/\beta))$ for $\beta > 0$, then with probability at least $1 - \beta$:

$$|\frac{1}{n}\sum_{i=1}^{n} f(\langle \Phi y_i, \Phi\theta\rangle, z_i) - \frac{1}{n}\sum_{i=1}^{n} f(\langle y_i, \theta\rangle, z_i)| \leq \gamma\|\mathcal{C}\|_2. \tag{11}$$

We will establish the connection of $\theta$ and $w^{\text{priv}}$ by the following lemma:

**Theorem 9.** [14] Let $\Phi$ be a random matrix in Theorem 7. Let $\mathcal{C}$ be a convex set. Given $v = \Phi u$, and let $\hat{u}$ be the solution to the following convex program: $\min_{u' \in \mathcal{R}^p} \|u'\|_\mathcal{C}$, subject to $\Phi u' = v$. Then for any $\beta > 0$, with probability at least $1 - \beta$,

$$\sup_{u:v=\Phi u} \|u - \hat{u}\|_2 \leq O(\frac{\psi^4 \mathcal{G}_\mathcal{C}}{\sqrt{m}} + \frac{\psi^4 \|\mathcal{C}\|_2 \sqrt{\log(1/\beta)}}{\sqrt{m}}). \tag{12}$$

Combing (9)(10)(11)(12) together, we have the following bound:

**Theorem 10.** Under the assumption above and setting $m = \Theta(\frac{\psi^4(\mathcal{G}_\mathcal{C} + \sqrt{\log n})^2 \log(n/\beta)}{\gamma^2})$ for $\gamma < 1$, then with probability at least $1 - \beta$,

$$\text{Err}_D(w^{\text{priv}}) = \tilde{O}((\frac{\log^2(1/\beta)\sqrt{m}}{n\epsilon^2})^{\frac{1}{1+m}} + \gamma), \tag{13}$$

where $\psi$ is the sub-Gaussian norm of the distribution of $\Phi$, and $\mathcal{G}_\mathcal{C}$ is the Gaussian width of $\mathcal{C}$.

Then taking $\gamma$ as in the Theorem, we have the proof.

For the corollary we will use the property that $\mathcal{G}_\mathcal{C} = O(\sqrt{\log p})$.

# F  Conclusion and Discussion

In this paper, we studied ERM under non-interactive LDP and proposed an algorithm which is based on Bernstein polynomial approximation. We showed that if the loss function is smooth enough, then the sample complexity to achieve $\alpha$ error is $\alpha^{-c}$ for some positive constant $c$, which improves

significantly on the previous result of $\alpha^{-(p+1)}$. Moreover, we proposed efficient algorithms for both player and server views. Our techniques can also be extended to high dimensional space and some other related problems to to answering k-way-marginals and smooth queries in the local model.

In our algorithms the sample complexity still depends on the dimension $p$, in the term of $c^p$ for constant $c$. We will focus on removing this dependency in future work. Additionally, we will study the difference between strongly convex and convex loss functions in the non-interactive LDP setting.