[Reviews · NeurIPS 2018]

Reviewer 1



Summary: This paper considers locally differentially private (LDP) protocols under the non-interactive setting. Specifically, they study the LDP empirical (and population) risk minimization problem. Existing work (Smith et al. [19]) states that in general alpha^{-p} term is unavoidable in the sample complexity of any LDP algorithm for this problem. One of the main contributions of this work is to show that it is indeed possible to avoid this term under certain smoothness conditions. For this, they have used the function approximation via Bernstein polynomials (similar to [1]). They also propose and analyze the sample complexity of a computation and communication variant of their initial solution (Algorithm 2->3). For this, they exploit the 'sampling resilient' property of the LDP-AVG algorithm. Comments: The paper addresses a significant problem, and it extends ideas/techniques from several existing works to answer the questions posed in the introduction of the article. In the beginning, the paper is well written - problem statement, motivation, and contributions are clearly stated. But the latter parts of sections 5 and 6 are very dense with technical details, with minimal/no discussions following the theorems. Minor typos: 'd' is used mistakenly in place of 'p' (e.g. line 232) Questions: 1) The resulting risk bounds of Corollaries 1 and 2 are compared to that of Theorem 1. But is it possible to compare them with the results mentioned in lines 61-68 (Smith et al. [19], Zheng et al. [24])? Are the loss functions mentioned there satisfy your smoothness conditions ((8,T), and (infinity,T))? If then how strong/weak is your bound compared to that? 2) Is the LDP-AVG protocol mentioned in Lemma 2, a multivariate variant of Algorithm 1 or is mentioned anywhere else explicitly?

Reviewer 2



- Summary: This paper considers the problem of empirical risk minimization in the non-interactive setting of the local model of differential privacy. In this setting, each user (holding one data point) is required to send a differentially private signal to the server without any prior interaction with the server or other users. Then, the server collects the users' signals and uses them to solve the ERM problem. The most relevant previous work is [19] that shows that any protocol that is based on first (or second) order methods (e.g., gradient descent and other variants) must require sample size \Omega(\alpha^{-p}) if it were to achieve error \alpha (where p is the dimensionality of the parameter space). This reference also gives upper bounds of the same order for non-interactive ERM under Local Differential Privacy (LDP) for the class of Lipschitz loss functions and the class of Lipschitz, convex loss functions. This paper revisits this problem under some smoothness assumptions on the loss function, and devises new algorithms for this problem based on polynomial approximation techniques. The goal is to improve the previous bounds for non-interactive ERM under LDP for some classes of smooth loss functions. When the sample size (n) < the dimensionality (p): despite success in showing improved sample complexity bounds, the resulting bounds still exhibit exponential dependence on p. For a sub-class of smooth functions (with bounded derivatives up to some constant order), the resulting bound is still ~ \alpha(-\Omega(p)), albeit with a better exponent than the one in [19] devised for Lipschitz loss functions. For a more restrictive class of smooth functions (with bounded derivative of any order), the resulting bound shows a better dependence on \alpha (namely, \alpha^{-4}), yet it still has from an extra factor that is exponential in p (about 4^{p^2}). When p > n: the paper gives an algorithm that results in sample complexity bounds that only depend (albeit exponentially) on the Gaussian width of the parameter set and n (when the loss function is a generalized linear function). The paper also shows improved constructions that has better running time and communication complexity than the ones in [19]. - Strengths: 1- The paper studies an important problem. The work is well-motivated. The introduction, as well as the outline of the paper and the high-level description of the algorithms and technical contributions, are nicely written. However, there are some parts in the technical sections that are not as clear (e.g., undefined notation, missing definitions), but these were not very critical (also these issues are understandable considering the space limitations). 2- The paper gives new algorithms for non-interactive ERM under LDP based on techniques from polynomial approximation. The paper also shows improved constructions in terms of running time and communication complexity based on some ideas from [2]. (however, there are some issues need to be addressed and clarified further (See "Weaknesses" below)). 3- Rigorous and original analyses. - Weaknesses: 1- Despite the strengths of the paper, the results are not quite impressive. The improvements in the bounds are hardly any useful even for p = 5, e.g., a sample complexity bound that scales with 4^{p^2} \alpha^{-4} is still not meaningful in this case (regardless of the fact that it's better than \alpha^{-2p} for some range of \alpha and p). 2- The paper raises an interesting question in the introduction and uses it to build up part of the motivation for this work, but it does not seem to address at all. The paper notes that in some special cases of loss functions (e.g., squared loss [the linear regression case in [19]] and logistic loss [24]) one can have bounds that are free from this exponential dependence on p in the non-interactive setting, and continues to say "this propels us the following natural questions .... Is it possible to introduce some natural conditions on the loss function that guarantee non-interactive _x000f_-LDP with sample complexity that is not exponential in the dimensionality p?". Although the class of smooth functions considered in this paper contains the two aforementioned examples of loss functions, there is no indication that the derived bounds can ever be free from this exponential dependence on p for those special cases of loss functions. 3- The bounds derived for generalized linear functions in terms of Gaussian width (for p > n case) are also hardly practical. Even if the Gaussian width is as small as log(p), the error bounds still scale roughly as ~ (1/n)^{1/\Omega(n)} (Corollary 3). Comparing this to known non-private nearly dimension-independent risk bounds for generalized linear models, it's not clear if the derived bounds are any useful. 4- There is an unquantified dependence on p in Corollary 2, Theorems 4 and 5 via the parameter D_p. The paper introduces D_p (once in the prelim section) as a "universal constant" that depends only on p. But, I don't see how it's a constant if it depends on p? Can this "constant" be at least upper and lower bounded by some truly universal constants independent of p? I think this dependence needs to be explained further. As stated, it's not clear if this D_p does not add an extra factor to the bound making it worse. 5- Some of the discussion in the paragraph after Corollary 2 is actually not clear. Why is it when h --> \infty, the dependence on the error becomes \alpha^{-2}? Isn't it alpha^{-4}? 6- In the analysis of Algorithm 3, the issue of shared/public randomness required in the setup needs to be further discussed. ------------------------------- ------------------------------- ****Post Rebuttal Comments: ----------------------------------- After reading the authors' rebuttal, I am inclined to keep my score as is. The authors agreed with several points made in my review, and they clarified most of the issues I had questions about. I think the work is theoretically interesting and removing the 1/\alpha from the base of the exponential dependence is nice. However, I still question the extent of the practicality of the main results especially with the prohibitive exponential dependence on the square of the dimensions in the final bounds. Also, the paper could not provide any insight for the reason why there are non-interactive schemes for some special cases of smooth losses (e.g, linear regression) that do not suffer from this exponential dependence at all.

Reviewer 3



When estimating statistics, learning models, and performing optimization on the private properties of a population, local-differential-privacy techniques have many advantages---notably, to protect their privacy, each respondent need not trust the data collector to keep their answers secret. For this type of distributed learning to be practical, it is important to minimize the sample complexity and rounds of interaction. This theory paper provides new methods and analysis for non-interactive distributed learning, establishing results that improve on the bounds from [19] with higher efficiency and wider applicability. However, for this a number of assumptions are required, in particular on smoothness of the loss. Overall score: This paper clearly makes advances in the theory of non-interactive local differential privacy, directly motivated by [19] and improving on their results. The paper provides new algorithms and sample-complexity bounds for smooth loss functions, shows how efficiency may be increased, as well as describing methods applicable in some high-dimensional cases. In addition, the paper is well written and structured. As such, its contributions clearly deserve publication. This said, the results are limited by assumptions about smoothness, and the impact of those limitations isn't clear, at least to this reviewer. Also, the paper contains no (empirical or analytic) evaluation of datasets, methods, or concrete applications that might shed light on the impact of the results. Confidence score: While this reviewer is familiar with but not expert in much of the related work, including [19]. The lack of empirical validation of the assumptions and application of the results make it hard to assess impact. Update: I thank the authors for their response the the reviews. I'm leaving my score as is, but I'll note that the paper would be much stronger if it more clearly offered substantial new insights or paths to practical benefits.